# Evaluation and comparison of MAIAC, DT and DB aerosol products over China

Ning Liu[1], Bin Zou[1,2], Huihui Feng[1], Wei Wang[1], Yuqi Tang[1], Yu Liang[1]

[1]School of Geosciences and Info-Physics, Central South University, Changsha, 410083, China

[2]Key Laboratory of Metallogenic Prediction of Nonferrous Metals and Geological Environment Monitoring(Central South University), Ministry of Education, Changsha, 410083, China

*Correspondence to*: Bin Zou (210010@csu.edu.cn);

**Abstract.** A new Multiangle Implementation of the Atmospheric Correction (MAIAC) algorithm has been applied in the
Moderate Resolution Imaging Spectroradiometer (MODIS) sensor and has recently provided globally high spatial resolution Aerosol Optical Depth (AOD) products at 1 km. Moreover, several improvements have been modified in the classical Dark Target (DT) and Deep Blue (DB) aerosol retrieval algorithms in MODIS collection 6.1 products. Thus, validation and comparison of the MAIAC, DT and DB algorithms are urgent in China. In this paper, we present a comprehensive assessment and comparison of AOD products at a 550 nm wavelength based on three aerosol retrieval algorithms in the MODIS sensor
using ground-truth measurements from Aerosol Robotic Network (AERONET) sites over China from 2000 to 2017. In general, MAIAC products achieved better accuracy than DT and DB products in the overall validation and accuracy improvement of DB products after the QA filter, demonstrating the highest values among the three products. In addition, the DT algorithms had higher aerosol retrievals in cropland, forest, and ocean land types than the other two products, and the MAIAC algorithms were more accurate in grassland, built-up, unoccupied land and mixed land types among the three products. In the geometry
dependency analysis, the solar zenith angle, scattering angle and relative azimuth angle, excluding the view zenith angle, significantly affected the performance of the three aerosol retrieval algorithms. The three products showed different accuracies with varying regions and seasons. Similar spatial patterns were found for the three products, but the MAIAC retrievals were smaller in the North China Plain and higher in Yunnan Province compared with the DT and DB retrievals before the QA filter. After the QA filter, the DB retrievals were significantly lower than the MAIAC retrievals in South China. Moreover, the
spatiotemporal completeness of the MAIAC product was also better than the DT and DB products.

## 1 Introduction

Aerosols are a multi-compartment system consisting of suspended solid and liquid particles in the atmosphere, which play an important role in radiative forcing (Rajeev et al., 2001), regional climate (Qian et al., 1999; Feng et al., 2019a) and urban air pollution (Dominici et al., 2014). The aerosol optical depth (AOD) is the key aerosol optical parameter, defined as the vertical
integration of the aerosol extinction coefficient from the ground to the top of the atmosphere (TOA). Ground measurements

from the Aerosol Robotic Network (AERONET) provide high-quality multiband aerosol optical and microphysical properties at 15-min sampling frequencies on a global scale (Holben et al., 2001). High-quality ground measurements are often employed to validate satellite aerosol products (Chu et al., 2002) and to provide a regional aerosol model for the satellite aerosol retrieval algorithm (Levy et al., 2013). However, they cannot grasp the high aerosol spatial variability due to the sparse ground sites

where spatial variability information is still necessary. Some active remote sensing methods, e.g. spaceborne lidar, can monitor vertical distribution of aerosol, they still cannot observe high aerosol spatial variability (Huang et al., 2007; Jia et al., 2015; Liu et al., 2015). Although model simulated AOD can spatial continuous data, its very coarse resolution and large uncertainties limit its application (Sun et al., 2018; Cesnulyte et al., 2014). In contrast, satellite aerosol retrieval algorithm has the ability to achieve continuous spatial measurements with high spatial resolution (She et al., 2017).

The Moderate Resolution Imaging Spectroradiometer (MODIS) sensor with its multiband detection ability from the visible band to thermal infrared spectrum band (Salomonson et al., 1989) can readily detect aerosol properties. With the Terra satellite and Aqua satellite carrying the MODIS sensor successfully launched in 2000 and 2002, respectively, it has stored over 17 years of globally historical monitored data. Recently, a new Multiangle Implementation of the Atmospheric Correction (MAIAC) algorithm has been applied in the MODIS sensor, which provides high spatial resolution aerosol data at 1 km

(Lyapustin et al., 2018). Moreover, some important improvements in classical Dark Target (DT, Mattoo et al., 2017) and Deep Blue (DB, Hsu, 2017) aerosol retrieval algorithms have been revised in MODIS collection 6.1 products. However, all satellite aerosol retrieval algorithms are under some hypothesis and approximation assessments, and the accuracy should be validated before applying a satellite aerosol product in related studies.

China is experiencing severe aerosol pollution, and numerous studies on aerosol pollution have utilized MODIS collection 6.0

aerosol retrievals to map aerosol pollution and to analyse its spatiotemporal trends (Fang et al., 2016; Ma et al., 2014; He et al., 2018; Zou et al., 2016, 2019; Zhai et al., 2018). Few studies have applied 1 km MAIAC aerosol retrievals to map finer aerosol concentrations in regional China, e.g., the Yangtze River Delta (Xiao et al., 2017) and Shandong Province (Li et al., 2018). Before widely applying MAIAC and C6.1 products in China, the accuracy differences and applicable conditions of the three aerosol retrievals should first be recognized to guide the utilization of these products. Recently, the global validation

(Lyapustin et al., 2018) and regional validation in South America (Martins et al., 2017), North America (Superczynski et al., 2017) and South Asia (Mhawish et al., 2019) for MAIAC products have shown that more than 66% of retrievals fall within the expected error (EE=±(0.05+0.05×AOD)) limits, indicating a good accuracy for MAIAC products. In China, a comprehensive validation of the C6.1 product were initially performed (Wang et al., 2019) and then the MAIAC product is relative simply evaluated against ground AERONET measurement in different seasons, land cover types and different sites

(Zhang et al., 2019). Thus, an urgent demand persists for a detailed comparison of the three products to guide user selection of these products.

In this context, we provide the first comprehensive understanding and comparison of the aerosol retrieval uncertainties for MAIAC, DT and DB products in China based on spatiotemporal accuracy differentiation patterns, spatiotemporal completeness, land type dependence characteristics, view geometry dependence characteristic aspects, and other features. The

following paper is organized as follows: section 2 briefly introduces three satellite products with their retrieval algorithm and ground AERONET data, the validation approach is clarified in section 3, and section 4 provides the detailed validation results and discussion. The conclusion are presented in section 5.

## 2 Data Description

Three aerosol products, e.g., MAIAC, DT and DB are stored in Hierarchical Data Format (*.hdf) files, and we obtain corresponding *.hdf files in the China region from 2000 to 2017 from the NASA Earthdata Search website (https://search.earthdata.nasa.gov/search/). Due to snow, cloud and desert ground surface types will increase the retrieval uncertainty, and three product provides a quality assurance (QA) flag to indicate the retrieval quality. Ground measurement aerosol data obtained from the AERONET website (https://aeronet.gsfc.nasa.gov/) were used to validate the accuracy of three
satellite aerosol products. Additionally, land cover data from the Geographical Information Monitoring Cloud Platform (GIMCP, http://www.dsac.cn/) were utilized to analyse the land cover dependency for three satellite aerosol products.

### 2.1 DT products

  The DT algorithm retrieves AOD parameters based on the assumption that the surface reflectance in two visible bands, e.g., 470 nm and 644 nm, presents a good linear relationship with the surface reflectance in the shortwave infrared (SWIR) band,
e.g., 2119 nm, in dark, dense vegetated area, and the measurement in SWIR band is transparent with the aerosol particle (Kaufman et al., 1997, Levy et al., 2013). The surface and aerosol information can then be decoupled from the TOA spectral reflectance. Compared with the DT algorithm in collection 6.0, the DT algorithm in collection 6.1 mainly revises the surface characterization over the land surface when the urban percentage is larger than 20% (Gupta et al., 2016).

  The DT algorithm produces two resolution aerosol products in collection 6.0 and 6.1, e.g., 3 km×3 km and 10 km×10 km. The
two resolution products share the same retrieval protocol except the use of different retrieval boxes. For example, the 10 km product organizes 20×20 group pixels with the three aforementioned band measurements at 500 resolution into the retrieval box, whereas the 3 km product combines three band measurements in the 6×6 pixel group into a retrieval box (Remer et al., 2013). The comparison between the 10 km product and 3 km product from collection 6.0 on the global scale (Remer et al., 2013) and the China region (He et al., 2017) shows that the accuracy of the 10 km product is superior to one of the 3 km
product, even though the 3 km product provides finer resolution aerosol retrievals. In this study, we consider the 10 km product of the newest collection 6.1 version from the Terra satellite. In DT products, QA=3 indicates high confidence data, and QA=1 indicates marginal confidence data (Levy, et al., 2013). In this paper, the scientific data set (SDS), named the "Image_Optical_Depth_Land_And_Ocean" QA filter and "Optical_Depth_Land_And_Ocean" with a QA filter (QA>1 for ocean and QA=3 for land) are extracted to compare the accuracy with and without the QA filter.

## 2.2 DB products

The DB algorithm retrieves the AOD parameter under the hypothesis that the surface reflectance in the deep blue band, e.g., 412 nm, is much smaller than in longer bands over bright surfaces, such as urban and desert regions (Hsu et al., 2004). First, the DB algorithm retrieves 1 km aerosol properties using the global surface reflectance database in visible bands, e.g., 412 nm,

470 nm and 650 nm, and then aggregates 1 km pixels into a 10 km scale. In collection 6.0, the surface reflectance database is improved using knowledge of the normalized difference vegetation index, scattering angle and season (Hsu et al., 2013). The ability to retrieve aerosol data over a bright surface for the DB algorithm greatly expands the coverage of aerosol retrieval. The general principles for collection of the 6.1 DB products are still the same as those in the collection 6.0 version. The major improvements for collection 6.1 DB products are in the radiometric calibration, heavy smoke detection, artefact reduction over

heterogeneous terrain, surface model in elevated terrain and regional/seasonal aerosol optical models (Hsu, 2017).

The same to the DT products, SDS named "Deep_Blue_Aerosol_Optical_Depth_550_Land" without the QA filter and "Deep_Blue_Aerosol_Optical_Depth_550_Land_Best_Estimate" with the QA filter (QA=2, 3 for land) in collection 6.1 from Terra satellite were selected for our study to validate the accuracy improvement by the QA filter. The solar zenith angle in "Solar_Zenith" SDS datasets, view zenith angle in "Sensor_Zenith" SDS datasets, solar azimuth angle in "Solar_Azimuth"

SDS datasets, sensor azimuth angle in "Sensor_Azimuth" SDS datasets and scattering angle in "Scattering_Angle" SDS datasets were also assessed to determine the geometry dependence for DT and DB products.

## 2.3 MAIAC products

The MAIAC algorithm relies on the assumption that the surface reflectance changes slowly over time and shows high variability over space, whereas the aerosol loading changes very fast over time and varies only on a limited space scale. The

main procedure of MAIAC is as follows: first, MAIAC resamples MODIS L1B measurements into a fixed 1 km grid, and then it adopts 4~16 day time series of resampled MODIS measurement to retrieve the surface Ross–Thick Li–Sparse (RTLS) bidirectional reflectance distribution function (Lucht et al., 2000) using the measurements in SWIR band. Subsequently, the linear spectral regression coefficient (SRC) between 470 nm and 2119 nm for each 1 km grid is retrieved instead of using the empirical regression coefficient in the DT algorithm. Finally, the AOD parameter at 470 nm can be retrieved by searching the

minimum spectral residual between the theoretical TOA reflectance of the look-up table and the measurements in the red and SWIR bands. The AOD is originally retrieved at 470 nm, and the AOD parameter at 550 nm is computed using the AOD parameter at 470 nm based on spectral properties, expressed by the regional aerosol model from the MAIAC look-up table. The detailed MAIAC algorithm has been described by Lyapustin et al., 2011.

Data used in this study were from the "Optical_Depth_055" and "AOD_QA" SDS datasets, and data were collected from the

Terra satellite. The datatype of the "AOD_QA" SDS datasets is a 16-bit unsigned integer, and the best retrieved quality can be selected if 8~11 bytes of "AOD_QA" SDS dataset bits is "0000", which indicates the retrieval pixel and its adjacent pixel is clear (Lyapustin et al., 2018). The solar zenith angle in the "cosSZA" SDS datasets, view zenith angle in the "cosVZA" SDS

datasets, relative azimuth angle in the "RelAZ" SDS datasets and scattering angle in the "Scattering_Angle" SDS datasets were also selected to analyse the view geometry dependence for MAIAC products.

## 2.4 AERONET data

AERONET is a global ground-based aerosol monitoring network that provides continuous optical and microphysical properties of aerosols at a 15-min sampling rate. The total uncertainty for the AERONET AOD parameter under cloud-free conditions is lower than $\pm 0.01$ for a wavelength longer than 440 nm and less than $\pm 0.02$ for shorter wavelengths (Holben et al., 1998). Some studies were also conducted to examine the properties of these high-quality measurements in China (Liu et al., 2011; Xia et al., 2007; Li et al., 2007; Che et al., 20018, 2014; Bi et al., 2014). These high-accuracy datasets support various satellite AOD product evaluation research in the China region (Tao et al., 2015; Tian et al., 2018; He et al., 2017; Sogacheva et al., 2018). AERONET provides three quality levels of data, e.g., level 1.0, level 1.5 and level 2.0, in version 3. Here, we only selected quality-assured level 2.0 data as ground-truth data to validate the satellite data. Figure 1 shows the locations of the selected 50 AERONET sites across China in this study. Table 1 reports the site name, longitude, and latitude of the selected sites. However, the AERONET site does not record aerosol measurements at 550 nm, and thus we interpolated the AOD parameter at 550 nm using the Ångström exponent in the two neighbouring bands at 500 nm and 675 nm (Ångström et al., 1929; Eck et al., 1999) which can be shown by

$$
\begin{aligned}
\alpha_{500-675} &= -\frac{\ln(\tau_{500}/\tau_{675})}{\ln(500/675)}, \\
\tau_{550} &= \tau_{675}(500/675)^{-\alpha_{500-675}},
\end{aligned}
\tag{1}
$$

where $\tau_{500}$ and $\tau_{675}$ are the AOD parameter at 500 nm and 675 nm, respectively, $\tau_{550}$ is the interpolated AOD parameter at 550 nm, $\alpha_{500-675}$ is the corresponding Ångström exponent and $\ln(*)$ is the logarithmic operator.

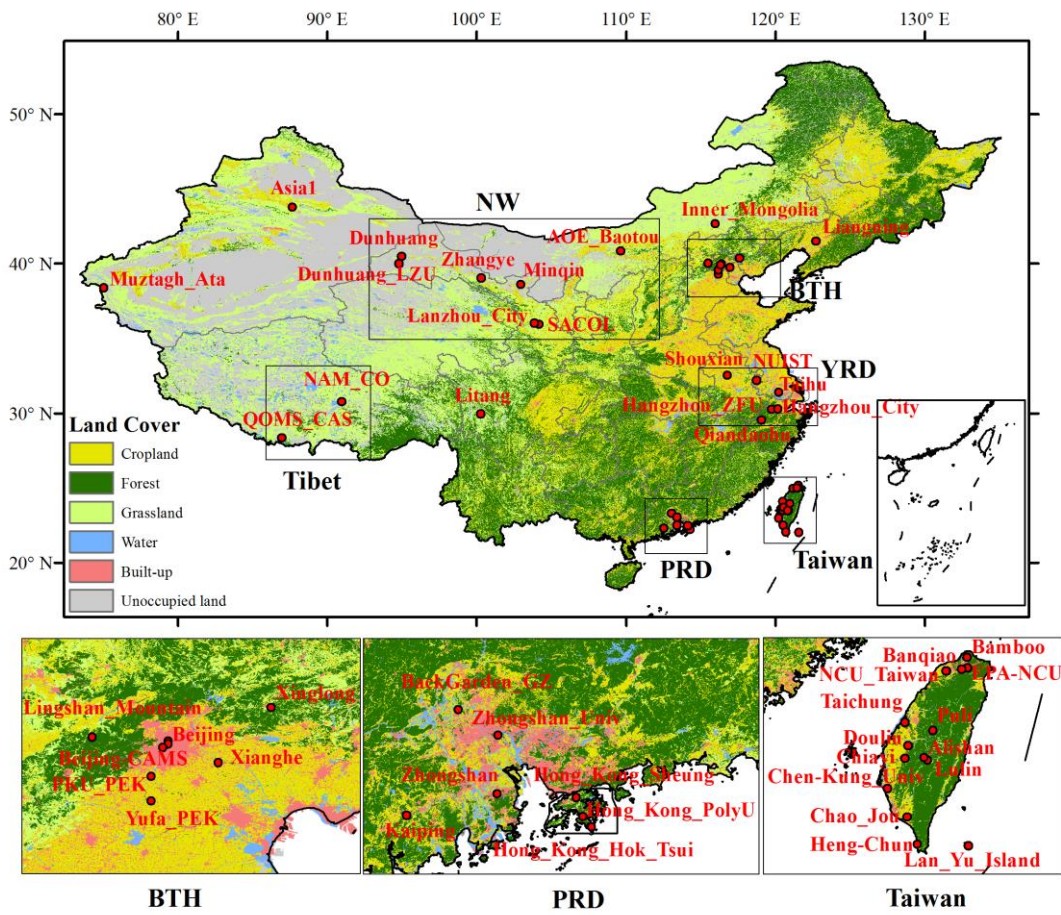

**Figure 1: Locations of the selected AERONET sites around China displayed on the land cover map from 2013. BTH: Beijing-Tianjin-Heibei; YRD: Yangtze River Delta; PRD: Pearl River Delta; NW: northwestern China.**

| Site | Longitude | Latitude | Period | Number of Match | | |
| --- | --- | --- | --- | --- | --- | --- |
| | | | | MAIAC | DT | DB |
| NCU_Taiwan | 121.19 | 24.97 | 1998-2013 | 100 | 94 | 96 |
| Taipei_CWB | 121.54 | 25.01 | 2000-2018 | 559 | 535 | 609 |
| Beijing | 116.38 | 39.98 | 2001-2018 | 527 | 207 | 371 |
| Dunhuang | 94.79 | 40.04 | 2001-2001 | 11 | 1 | 12 |
| Inner_Mongolia | 115.95 | 42.68 | 2001-2001 | 9 | 1 | 8 |
| Lan_Yu_Island | 121.56 | 22.04 | 2001-2001 | 9 | 4 | 0 |
| XiangHe | 116.96 | 39.75 | 2001-2018 | 3485 | 1841 | 2267 |
| Chen-Kung_Univ | 120.20 | 22.99 | 2002-2018 | 593 | 553 | 623 |
| EPA-NCU | 121.19 | 24.97 | 2004-2018 | 411 | 373 | 394 |
| Chao_Jou | 120.53 | 22.51 | 2005-2005 | 4 | 3 | 3 |
| Hong_Kong_PolyU | 114.18 | 22.30 | 2005-2018 | 661 | 423 | 497 |

| | | | | | | |
|---|---|---|---|---|---|---|
| Liangning | 122.7 | 41.51 | 2005-2005 | 13 | 26 | 27 |
| Taichung | 120.49 | 24.11 | 2005-2005 | 16 | 16 | 16 |
| Taihu | 120.22 | 31.42 | 2005-2018 | 286 | 413 | 595 |
| BackGarden_GZ | 113.02 | 23.30 | 2006-2006 | 3 | 3 | 4 |
| Lulin | 120.87 | 23.47 | 2006-2018 | 505 | 445 | 505 |
| NAM_CO | 90.96 | 90.96 | 2006-2018 | 413 | 71 | 164 |
| PKU_PEK | 116.18 | 39.59 | 2006-2008 | 23 | 16 | 16 |
| SACOL | 104.14 | 35.95 | 2006-2013 | 259 | 230 | 318 |
| Xinglong | 117.58 | 40.40 | 2006-2014 | 147 | 124 | 141 |
| Yufa_PEK | 116.18 | 39.31 | 2006-2006 | 11 | 9 | 9 |
| Asia1 | 87.65 | 43.78 | 2007-2007 | 2 | 1 | 1 |
| Hangzhou-ZFU | 119.73 | 30.26 | 2007-2009 | 5 | 22 | 22 |
| Hong_Kong_Hok_Tsui | 114.26 | 22.21 | 2007-2010 | 120 | 62 | 78 |
| NUIST | 118.72 | 32.21 | 2007-2010 | 7 | 5 | 9 |
| Qiandaohu | 119.05 | 29.56 | 2007-2009 | 53 | 49 | 53 |
| Hangzhou_City | 120.16 | 30.29 | 2008-2009 | 29 | 43 | 59 |
| Kaiping | 112.54 | 22.32 | 2008-2008 | 10 | 9 | 9 |
| Shouxian | 116.78 | 32.56 | 2008-2008 | 37 | 42 | 48 |
| Zhangye | 100.28 | 39.08 | 2008-2008 | 37 | 22 | 27 |
| Lanzhou_City | 103.85 | 36.05 | 2009-2010 | 27 | 19 | 45 |
| QOMS_CAS | 86.95 | 28.37 | 2009-2018 | 1241 | 36 | 715 |
| Zhongshan | 113.38 | 22.52 | 2009-2009 | 2 | 2 | 2 |
| Beijing_RADI | 116.38 | 40.00 | 2010-2018 | 116 | 68 | 127 |
| Minqin | 102.96 | 38.61 | 2010-2010 | 15 | 8 | 18 |
| Litang | 100.26 | 29.98 | 2011-2011 | 2 | 2 | 2 |
| Muztagh_Ata | 75.04 | 38.41 | 2011-2011 | 104 | 0 | 41 |
| Zhongshan_Univ | 113.39 | 23.06 | 2011-2012 | 34 | 28 | 30 |
| Beijing-CAMS | 116.32 | 39.93 | 2012-2018 | 1250 | 505 | 818 |
| Dunhuang_LZU | 94.96 | 40.49 | 2012-2012 | 17 | 0 | 22 |
| Hong_Kong_Sheung | 114.12 | 22.48 | 2012-2018 | 94 | 67 | 81 |
| AOE_Baotou | 109.63 | 40.85 | 2013-2018 | 16 | 21 | 24 |
| Chiayi | 120.5 | 23.5 | 2013-2018 | 287 | 289 | 298 |
| Heng-Chun | 120.7 | 22.05 | 2013-2015 | 59 | 35 | 36 |
| Puli | 120.97 | 23.97 | 2013-2013 | 4 | 3 | 3 |
| Lingshan_Mountain | 115.5 | 40.05 | 2014-2015 | 1 | 1 | 1 |
| Douliu | 120.54 | 23.71 | 2015-2018 | 70 | 69 | 72 |
| Alishan | 120.81 | 23.51 | 2016-2016 | 5 | 4 | 4 |
| Bamboo | 121.54 | 25.19 | 2016-2017 | 1 | 1 | 1 |
| Banqiao | 121.44 | 25.00 | 2017-2017 | 26 | 20 | 22 |

**Table 1: Selected AERONET sites used in this study. The number of the match column statistics match the number between the satellite observations before the QA filter and the ground AERONET observations in the selected spatiotemporal window presented in section 3.1.**

**2.5 Land cover data**

One key difficulty in the aerosol retrieval algorithm is to decouple surface and atmosphere information in the satellite apparent reflectance. Land cover information greatly affects atmosphere properties (Xu et al., 2018; Feng et al., 2019b). Understanding the uncertainties in a satellite aerosol retrieval algorithm for different land cover types is necessary. GIMCP land cover data with 30 m resolution in the years 2000, 2005, 2008, 2010, and 2013 were used in this study. The first level of GIMCP land cover data includes cropland, forest, grassland, water, built-up and unoccupied land. Among them, unoccupied land includes desert, gobi, saline-alkali soil, swampland, bare land and bare rock gravel, which mainly includes bright surfaces. The high spatial resolution and abundant land cover types support our studies. Figure 1 shows the first level land cover type across the China mainland in 2013.

**3. Evaluation method**

**3.1 The selected spatiotemporal window**

There is only a small amount of matchup data between the satellite data and ground data when using the direct matching method, e.g., use only one pixel where the AERONET sites is located and ground measurement at the exact satellite overpass time, due to large amounts of missing data in AERONET or satellite data and the time delay between the satellite overpass time and AERONET sampling time. Therefore, under the assumption that aerosol information is homogeneous in a limited spatial and temporal lag, a suitable spatiotemporal window is often adopted to increase the matchup data number. Thus, satellite measurements in the spatial window around the AERONET sites are averaged, and ground measurements in the temporal window centred on the satellite overpass time are averaged.

For 10 km DT and DB products, the selected spatial window is often 50 km×50 km, and the temporal window is ±30 min (Ichoku et al., 2002, He et al., 2017, Tao et al., 2015). For MAIAC products, Matins et al. described five different spatial windows, e.g., 3 km, 15 km, 25 km, 75 km and 125 km, and four temporal windows, e.g., 30 min, 60 min, 90 min and 120 min, to validate the MAIAC product over South America (Matins et al., 2017). The results showed that 25 km×25 km and ±60 min are reasonable for the Terra satellite. For comparison with 10 km DT and DB products, we selected 30 km×30 km as the spatial window closest to the best spatial window for the MAIAC product and employed the best temporal window ±30 min of 10 km product because we also noticed that the validation accuracy is very close for ±30 min and ±60 min temporal window in the results of Matins et al., although the matchup data number of the ±60 min temporal window is more than one of the ±30 min temporal window (Matins et al., 2017).

**3.2 Land cover types for the AERONET sites**

The first level of GIMCP land cover data were used to label the AERONET site group. Due to the selected 30 km×30 km spatial window in section 3.1, we labelled the AERONET sites if the proportion of one land cover type exceeded 50% in the

spatial window around the AERONET site. If there was no dominant land cover type, we defined the land cover type for this AERONET site as a mixed group. Except for the defined first level type in the GIMCP land cover data, we found some coastal AERONET sites in which the dominant region was ocean, so we defined the land cover type for these sites as the ocean group. Table 2 shows the land cover types for each AERONET site in 2013. There were no land cover type changes for most sites

except Hangzhou_City, Muztagh_Ata and NAM_CO site. For the Hangzhou_City site, the land cover type changed from cropland to mixed group from 2005 to 2008, potentially due to the process of urbanization. For the Muztagh_Ata site, the land cover type changed from unoccupied land to grass land from 2008 to 2010, and the land cover type for the NAM_CO site varied from grassland to the mixed group between 2008 and 2010. We labelled each matchup dataset for the three sites using the land cover type in the nearest year to the AERONET sampling time.

| Land Cover | Site | Land Cover | Site | Land Cover | Site |
| --- | --- | --- | --- | --- | --- |
| Cropland | Shouxian | Grassland | SACOL | Mixed | NCU_Taiwan |
| | XiangHe | | Asia1 | | Chen-Kung_Univ |
| | Liangning | | Lanzhou_City | | EPA-NCU |
| | PKU_PEK | | QOMS_CAS | | Chao_Jou |
| | Yufa_PEK | | Litang | | Taichung |
| | NUIST | | Muztagh_Ata | | Taihu |
| Forest | Taipei_CWB | | AOE_Baotou | | BackGarden_GZ |
| | Lulin | Built-up | Beijing | | NAM_CO |
| | Xinglong | | Beijing_RADI | | Hangzhou_City |
| | Hangzhou-ZFU | | Beijing-CAMS | | Kaiping |
| | Qiandaohu | Ocean | Lan_Yu_Island | | Zhangye |
| | Chiayi | | Hong_Kong_PolyU | | Zhongshan |
| | Puli | | Hong_Kong_Hok_Tsui | | Zhongshan_Univ |
| | Lingshan_Mountain | | Heng-Chun | | Hong_Kong_Sheung |
| | Alishan | Unoccupied | Dunhuang | | Douliu |
| | Banqiao | land | Minqin | | Bamboo |
| Grassland | Inner_Mongolia | | Dunhuang_LZU | | |

**Table 2: Land cover type for each AEROENT site in 2013.**

**3.3 Statistical approach**

The expected error (EE) envelope is often used to validate satellite retrieval uncertainties. More than 66% of retrievals falling within the expected error lines indicates good accuracy. For the DT algorithm, the EE envelope is generally defined as $\pm(0.05+0.15\times AOD)$ over land, and over 66% of retrievals meet the defined expected error limits at the global scale (Levy et

al., 2010, 2013; Remer et al., 2005). In the global scale validation for MAIAC product, over 66% of retrievals satisfy the

±(0.05+0.1×AOD) EE limits, demonstrating that the accuracy of MAIAC is relatively higher than the DT algorithm over land (Lyapustin et al., 2018). In the regional validation of South America and South Asia for the MAIAC product, the EE envelope is defined as ±(0.05+0.05×AOD) and ±(0.05+0.1×AOD) respectively, and the fraction of retrievals within these EE limits are all over 66% (Mhawish et al., 2019; Matins et al., 2017). In our study, to compare DT and DB products, we adopted

±(0.05+0.15×AOD) as the EE envelope and calculated the proportion within the EE envelope (Within_EE) using equation (2):

$$AOD - EE \leq AOD_{sat} \leq AOD + EE, \tag{2}$$

In addition to the EE envelope, we also adopted coefficient of determination ($R^2$) and Pearson correlation coefficient (R) to study the correlation between the satellite AOD and AERONET AOD. The root mean square error (RMSE) was also utilized to analyse the dispersion degree of accuracy of the satellite AOD. The mean bias (Bias) was used to describe the bias of the

satellite AOD. These statistical indicators were calculated using equation (3)-(6), respectively.

$$R^2 = 1 - \frac{\sum_{i=1}^{N}(AOD_{sat}-AOD_{aero})^2}{\sum_{i=1}^{N}(AOD_{aero}-\overline{AOD}_{aero})^2}, \tag{3}$$

$$R = \frac{\sum_{i=1}^{N}(AOD_{sat}-\overline{AOD}_{sat})(AOD_{aero}-\overline{AOD}_{aero})}{\sqrt{\sum_{i=1}^{N}(AOD_{sat}-\overline{AOD}_{sat})^2 \sum_{i=1}^{N}(AOD_{aero}-\overline{AOD}_{aero})^2}}, \tag{4}$$

$$RMSE = \sqrt{\frac{\sum_{i=1}^{N}(AOD_{sat}-AOD_{aero})^2}{N}}, \tag{5}$$

$$Bias = \frac{\sum_{i=1}^{N}(AOD_{sat}-AOD_{aero})}{N}, \tag{6}$$

The $AOD_{sat}$ and $AOD_{aero}$ are the satellite AOD retrievals and AERONET data, respectively. The $\overline{AOD}_{sat}$ and $\overline{AOD}_{aero}$ are the corresponding mean values. $N$ is the matchup data number.

In order to compare the spatiotemporal completeness of three products, daily spatial completeness and the temporal completeness are defined by equation (7)-(8).

$$Spatial\ Completeness = \frac{availble\ AOD\ pixel\ numbers}{the\ total\ number\ of\ pixels\ in\ the\ study\ region} \times 100\%, \tag{7}$$

$$Temporal\ Completeness = \frac{availble\ AOD\ numbers\ in\ each\ pixel\ during\ the\ study\ period}{The\ length\ of\ the\ study\ period} \times 100\%, \tag{8}$$

All the statistical indicators are calculated for three products before and after QA filter to indicate the accuracy improvements and the reduction of spatiotemporal completeness by QA flag.

## 4. Results and Discussion

### 4.1 Overall accuracy comparison

Figure 2 shows the overall evaluation for MAIAC, DT and DB products before and after the QA filter. In total, MAIAC products have more matchup data than DT and DB products, which indicates the completeness of the MAIAC product may be

higher than the DT and DB products. Before the QA filter, the statistic showed that 69.84% of retrievals fall within the EE envelope, indicating a good accuracy for MAIAC products in China. Compared with DT and DB products, only 53.64% and 55.66% of retrievals were determined for DT and DB products. Based on the R statistical result, the results for the three products were all greater than 0.9, indicating that the three products are all well correlated with the ground-truth AERONET data. By contrast, the $R^2$ statistical result for MAIAC products, e.g., 0.847, was superior to those for the DT and DB products,

e.g., less than 0.8. From the $Bias$ statistical result, no significant bias was observed for the overall MAIAC product. However, according to the corresponding bias boxplot in different AOD bins, a slight overestimation was observed when the AODs were less than 0.5, and a slight underestimation when the AODs were between 0.5 and 1. The DT and DB products appeared to be less overestimated based on the $Bias$ result. From the corresponding bias boxplot, the mean bias result for each different AOD bin was also almost greater than zero. After the QA filter, the correlation for the MAIAC product slightly improved, but the

Within_EE result was slightly reduced, and the RMSE and $Bias$ results increased. From the corresponding bias boxplot subfigure, the positive mean biases when the AODs were less than 0.2 increased compared with corresponding results before the QA filter, and the negative biases when the AODs were between 0.5 and 1 were reduced. These phenomena resulted in the reduced overall accuracy. The reason for the changes in these statistical indicators will be explained in section 4.2. For the DT and DB products, the overall accuracies were all improved after the QA filter. The improvement of the DB product was more

obvious than one for the DT product. The Within_EE result was improved from 57.66% to 63.32%, and the mean biases in the bias boxplot showed no obvious overestimation trend after the QA filter. However, the DT product was still overestimated after the QA filter, and only a little improvement was achieved in the Within_EE result.

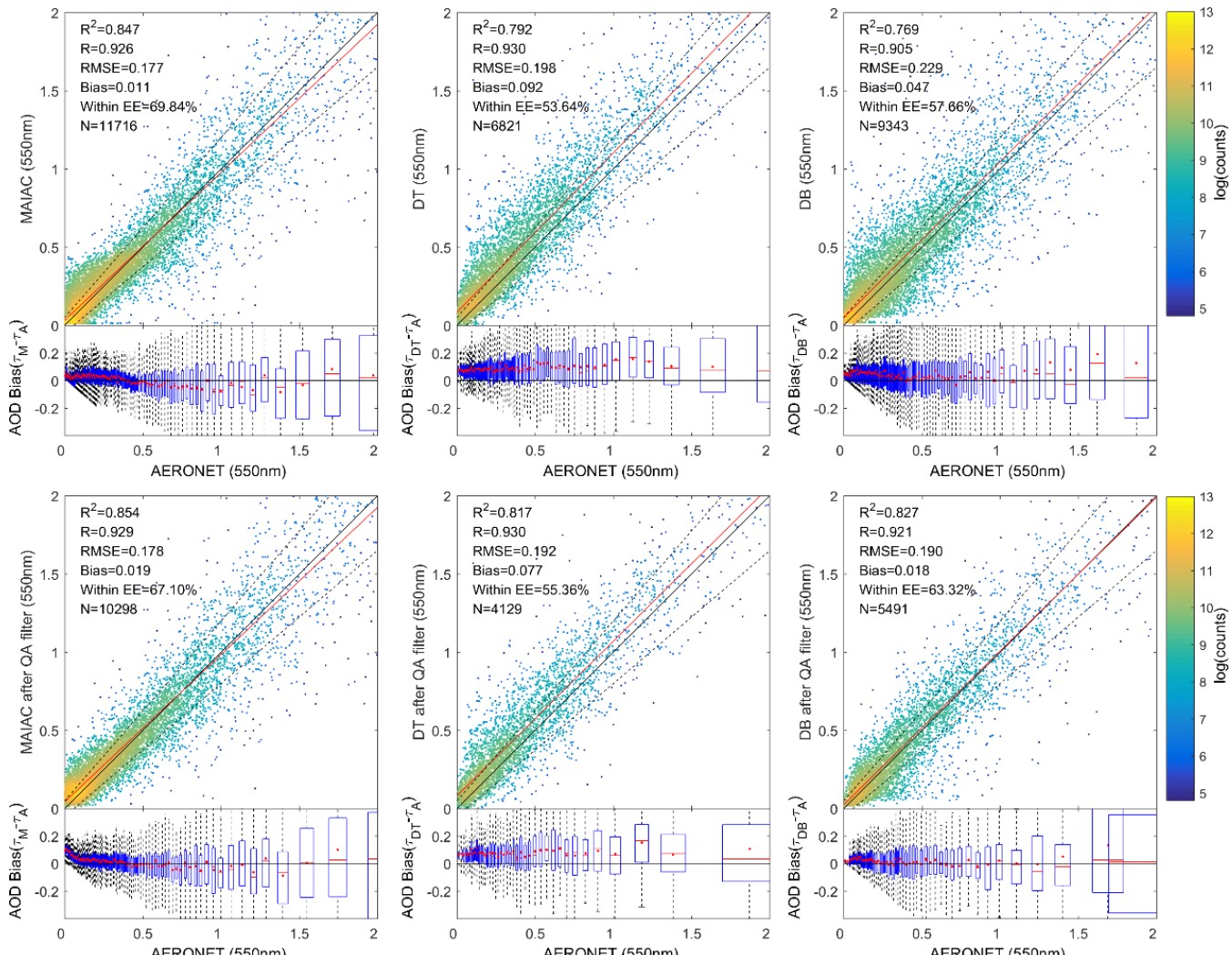

**Figure 2: Overall accuracy evaluation of MAIAC, DT and DB AOD versus AERONET AOD at 550 nm before and after the QA filter. The black line, red line and dashed line in the scatterplot are the 1:1 reference line, regression line and expected error (EE=±(0.05+0.15×AOD)) line, respectively. The matchup pairs are separated into 100 bins along with AERONET AOD values. The AOD bias boxplot uses the 25% and 75% percentiles for each 100 bins. The red points in the boxplot are the mean bias for each 100 bins.**

To analyse and compare the retrieval accuracy at different AOD levels for three products, four bins with different levels: low level (<0.2), moderate level (0.2~0.4), moderate-high level (0.4~0.6) and high level (>0.6) are defined (Wang et al. 2019). Table 3 shows the corresponding statistical results. At the low, moderate and moderate-high levels, all statistical indicators showed that the MAIAC product had better accuracy than the DT and DB products before the QA filter. At the high level, the DT product achieved the highest correlation with the ground-truth data and low RMSE results, but the positive bias result for the DT product revealed that the overestimation phenomenon was more serious than for the other two products. After the QA filter, the accuracy of the DB product was higher compared with the other two products at the low level because the positive

bias phenomenon became more severe for the MAIAC product at this level. At the moderate level, the MAIAC product demonstrated the best correlation and lowest RMSE results with a slightly higher positive bias than the DB product. At the moderate-high level, the MAIAC product remained the best quality product among the three products. At the high level, the DT product achieved the best correlation and lowest RMSE with the highest positive bias.

| AOD level | Data | Before QA filter | | | | After QA filter | | | |
|---|---|---|---|---|---|---|---|---|---|
| | | NOM | Bias | R | RMSE | NOM | Bias | R | RMSE |
| <0.2 | MAIAC | **5541** | **0.032** | **0.580** | **0.086** | **4521** | 0.047 | 0.435 | 0.084 |
| | DT | 2554 | 0.079 | 0.469 | 0.135 | 1478 | 0.077 | 0.455 | 0.137 |
| | DB | 3777 | 0.057 | 0.464 | 0.127 | 2090 | **0.031** | **0.555** | **0.082** |
| 0.2~0.4 | MAIAC | **2509** | **0.021** | **0.480** | **0.091** | **2320** | 0.016 | **0.501** | **0.091** |
| | DT | 1697 | 0.086 | 0.386 | 0.165 | 1038 | 0.070 | 0.361 | 0.159 |
| | DB | 2099 | 0.039 | 0.271 | 0.172 | 1361 | **0.008** | 0.408 | 0.128 |
| 0.4~0.6 | MAIAC | **1304** | **-0.017** | **0.396** | **0.129** | **1204** | **-0.007** | **0.421** | **0.132** |
| | DT | 989 | 0.105 | 0.394 | 0.202 | 605 | 0.081 | 0.388 | 0.188 |
| | DB | 1249 | 0.024 | 0.308 | 0.218 | 744 | 0.012 | 0.360 | 0.169 |
| >0.6 | MAIAC | **2362** | **-0.033** | 0.834 | 0.346 | **2253** | -0.019 | 0.840 | 0.336 |
| | DT | 1581 | 0.109 | **0.871** | **0.292** | 1008 | 0.081 | **0.876** | **0.277** |
| | DB | 2218 | 0.050 | 0.825 | 0.371 | 1296 | **0.010** | 0.836 | 0.330 |

**Table 3: Accuracy evaluation of MAIAC, DT and DB at the low level (<0.2), moderate level (0.2~0.4), moderate-high level (0.4~0.6) and high level (>0.6). NOM is the abbreviation for Number of Matches.**

**4.2 Land cover type dependency analysis**

Figure 3 shows a scatterplot figure of the MAIAC products in different land cover types before and after the QA filter. In total, MAIAC retrievals in cropland, built-up, grassland, ocean types were more accurate than forest, unoccupied land and mixed types according to the Within_EE results. After the QA filter, except for grassland, the accuracies all improved, and the improvement effect in ocean type was more obvious.

The high aerosol loading, e.g., AODs > 1, mostly emerged in cropland (Figure 3 a-i and a-ii) and built-up (Figure 3 d-i and d-ii) types due to biomass burning in the dry season and multiple human activities in the built-up area (Zhang et al., 2010; Wang et al., 2018). MAIAC retrieved AODs with a very high accuracy for the two land cover types. The R and $R^2$ results were over 0.93 and 0.84, respectively, and the Winthin_EE results showed that more than 74% of retrievals fell within the EE limits. In comparison, retrievals in cropland showed little bias, in contrast to a small positive bias in the built-up area, and RMSE results in the built-up area were smaller than those in the cropland area. This high retrieval accuracy in cropland and built-up regions can support relative studies on biomass burning and anthropogenic emissions.

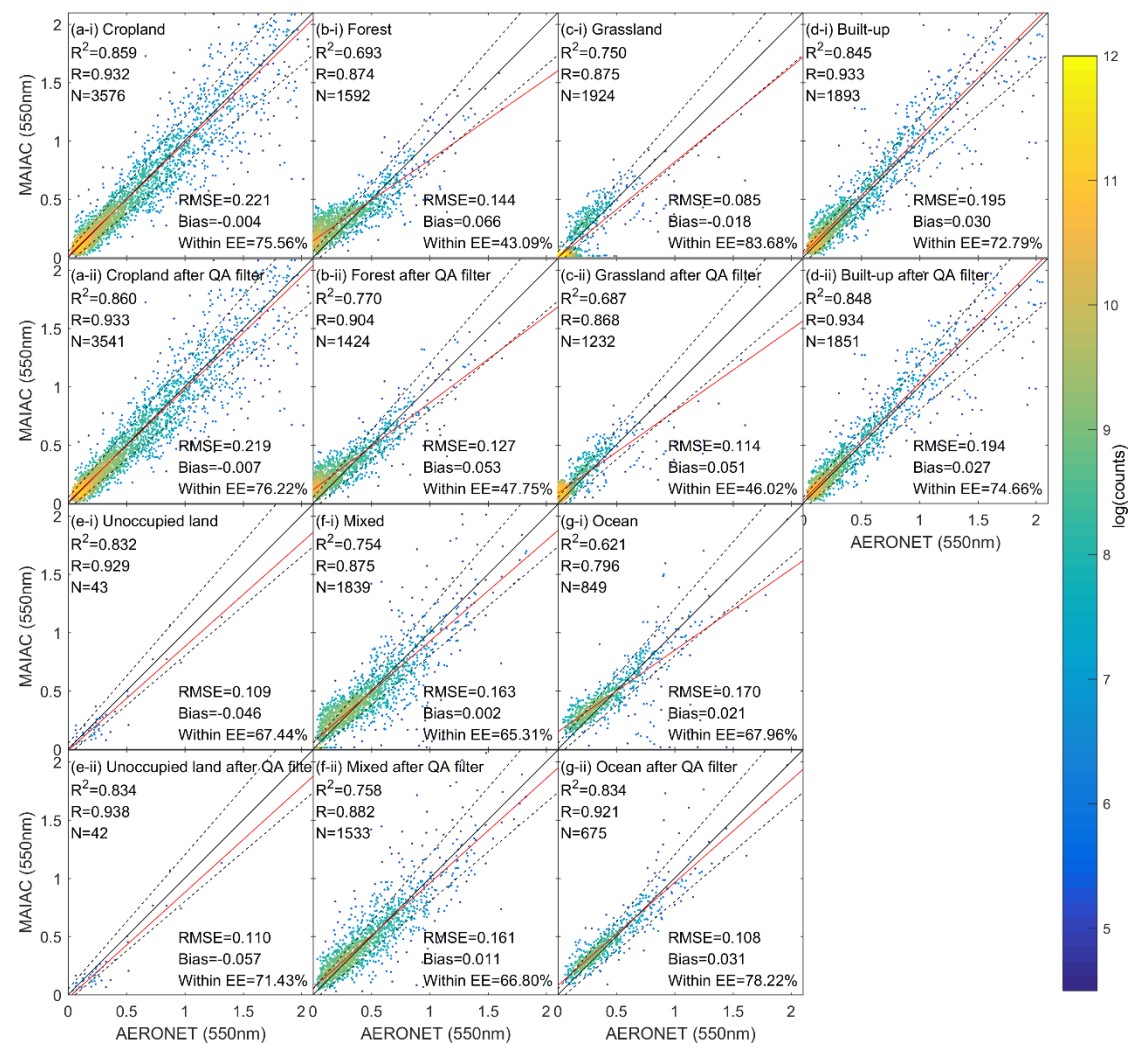

**Figure 3. Evaluation of the MAIAC accuracy for different land cover types before and after the QA filter. The black line, red line and dashed line in the scatterplot are the 1:1 reference line, regression line and expected error (EE=±(0.05+0.15×AOD)) line, respectively.**

In evergreen forest areas (Figure 3 b-i and b-ii), the retrievals showed a good correlation with ground measurements, with $R_{no\_QA} = 0.874$, $R_{QA} = 0.904$. However, the $R^2$ results without and with the QA filter were all lower than 0.8, and only approximately 45% of retrievals fell within the EE envelope. The result is opposite to the conclusion that the MAIAC algorithm improves the dark target retrieval accuracy better than the DT algorithm in Lyapustin et al., 2011 (Lyapustin et al., 2011). To eliminate the influence of retrieval accuracy in the specific site, Figure 4 shows a scatterplot figure of the forest AERONET site, ignoring the sites with matchup numbers less than 10. We can observe good performance in the Chiayi, Qiandaohu and Xinglong sites, and the corresponding Within_EE results were all higher than 70%. The relatively inferior performance sites were Banqiao, Taipei_CWB. After the QA filter, the accuracies were improved to 76.19% and 61.79%, respectively. The site with the worst performance was only the Lulin site, where the MAIAC retrievals were systemically higher than the ground

measurements, and less than 4% of retrievals fell within the EE limits. The percentage of forest type in the 30 km×30 km spatial window around the Lulin site always exceeded 80% in 2000, 2005, 2008, 2010 and 2013. This high proportion of forest type eliminates the influence of other mixed land cover type. The Lulin site is located in Taiwan peninsula, and thus the improper aerosol type in the MAIAC algorithm and cloud cover may explain the overestimation in the Lulin site (Lyapustin et al., 2018).

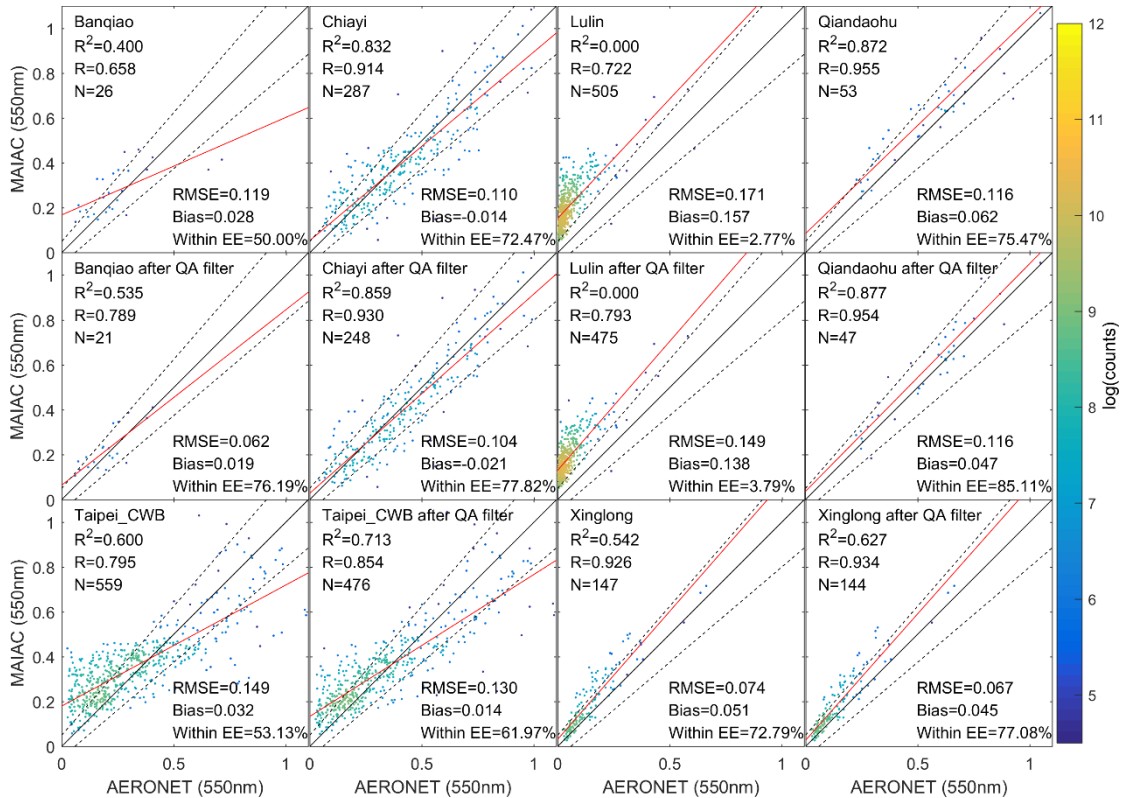

**Figure 4. Evaluation of the MAIAC accuracy in the forest area for each AERONET site before and after the QA filter. The black line, red line and dashed line in the scatterplot are the 1:1 reference line, regression line and expected error (EE=±(0.05+0.15×AOD)) line, respectively.**

10    For the grassland type (Figure 3 c-i and c-ii), over 83.68% of MAIAC retrievals fell into the EE lines before the QA filter, and the $R^2$=0.750, $R$=0.875, RMSE=0.085, and Bias=-0.018 results all showed good accuracy in the grassland type. However, after the QA filter, the accuracy was dramatically decreased with Within_EE=46.02%, $R^2$=0.687, $R$=0.868, Bias = 0.051 and RMSE=0.114, representing the main reason for some of the decreased overall statistical results shown in Figure 2 for the MAIAC product after the QA filter. It is noteworthy that some values were underestimated when the AODs were less than 0.5,

15    and these values were discarded after the QA filter. However, some overestimated values emerged when the AODs were very small. To identify the reason, we also performed a statistical validation for each grassland type site in Figure 5, excluding the site with a matchup number less than 10. Before the QA filter, the underestimated values were mainly in the NAM_CO and QOMS_CAS sites. These two sites are located in the Tibetan plateau. The MAIAC algorithm filled the AOD retrievals using

climatology values, e.g., 0.014, in high altitude regions, e.g., over 4.2 km, and the QA for climatology values was 0111 (Lyapustin et al., 2018). After the QA filter, the climatology values were thrown away in the NAM_CO site. For the QOMS_CAS site, nearly 2.13% of pixels still had altitudes less than 4.2 km in the spatial window. MAIAC retrievals in these pixels were overestimated compared with the ground measurements. After the QA filter, the Within_EE results decreased from 92.26% to 38.53%. A severe underestimation phenomenon was found in Lanzhou_City site, in contrast to the positive bias in its closest SACOL site. The small matchup number for the Lanzhou_City site might be the reason for the underestimation phenomenon. A great improvement was found in the Muztagh_Ata site after the QA filter.

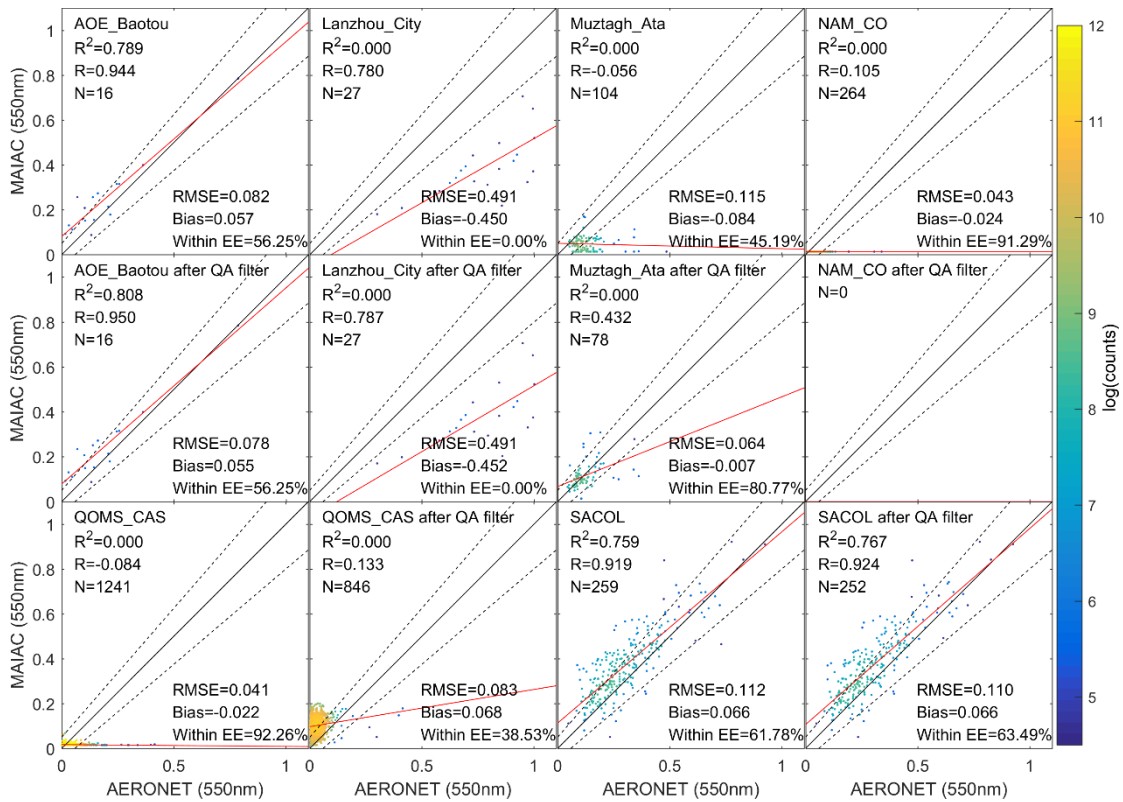

**Figure 5. Evaluation of the MAIAC accuracy in the grassland area for each AERONET site before and after the QA filter. The black line, red line and dashed line in the scatterplot are the 1:1 reference line, regression line and expected error (EE=±(0.05+0.15×AOD)) line, respectively.**

MAIAC had good accuracy in unoccupied land cover type (Figure 3 e-i and e-ii), with Within_EE results of 67.44% and 71.43% before and after the QA filter, and R and $R^2$ results over 0.9 and 0.8, respectively. Figure 3 f-i and f-ii indicate that MAIAC also achieved better performance in the mixed land cover area, with Within_EE=66.80% and R=0.882. In the ocean area (Figure 3 f-i and f-ii), MAIAC algorithm retrievals seemed to be overestimated when AODs were small, and the R=0.796 result was a little worse than one of the other land types. After the QA filter, the overestimated values were discarded, and the accuracy was greatly improved from R=0.796, Within_EE=67.96% to R=0.921, and Within_EE=78.22%.

In comparison to DT and DB products, Table 4 shows the validation of the statistical results for the MAIAC, DT and DB products with different land type covers. In cropland area, the accuracy of the DT product was evidently better than that of the MAIAC and DB products according to the $R^2$, R, and RMSE results. However, the values seemed to be overestimated compared with the MAIAC product, and the Within_EE result was a little smaller compared with the MAIAC product. In the forest area, the DT algorithm also achieved optimal accuracy compared with the MAIAC and DB products. However, only 56.23% of the retrievals met the EE limits, which was less than the DB product. In the grassland type region, the accuracies for the three products were all decreased after the QA filter, and we consider the validations of the three products to have all been influenced by the NAM_CO and QOMS_CAS sites. Compared with DT and DB products, the MAIAC product obtained the best retrieval accuracy. Owing to the overestimation phenomenon in the QOMS_CAS sites after the QA filter, the Within_EE result dramatically dropped from 83.68% to 46.02%. In Built-up, unoccupied land and mixed regions, the MAIAC product performed better than the DB product, and the DB product was more accurate than the DT product. In the ocean region, the DT product was clearly more accurate than the DB and MAIAC products.

| | | | Cropland | Forest | Grassland | Built-up | Unoccupied land | Mixed | Ocean |
|---|---|---|---|---|---|---|---|---|---|
| $R^2$ | Before QA filter | MAIAC | 0.859 | 0.693 | **0.750** | **0.845** | **0.832** | **0.754** | 0.621 |
| | | DT | **0.903** | **0.798** | 0.370 | 0.696 | ------- | 0.520 | **0.876** |
| | | DB | 0.813 | 0.636 | 0.550 | 0.799 | 0.428 | 0.600 | 0.434 |
| | After QA filter | MAIAC | 0.860 | 0.770 | **0.687** | **0.848** | **0.834** | **0.758** | 0.834 |
| | | DT | **0.915** | **0.812** | 0.038 | 0.579 | ------- | 0.553 | **0.838** |
| | | DB | 0.843 | 0.804 | 0.480 | 0.852 | 0.710 | 0.724 | 0.152 |
| R | Before QA filter | MAIAC | 0.932 | 0.874 | **0.875** | **0.933** | **0.929** | 0.875 | 0.796 |
| | | DT | **0.964** | **0.896** | 0.726 | 0.934 | ------- | **0.898** | **0.939** |
| | | DB | 0.927 | 0.850 | 0.744 | 0.928 | 0.689 | 0.832 | 0.777 |
| | After QA filter | MAIAC | 0.933 | 0.904 | **0.868** | **0.934** | **0.938** | **0.882** | 0.921 |
| | | DT | **0.966** | **0.901** | 0.585 | 0.916 | ------- | 0.875 | **0.941** |
| | | DB | 0.933 | 0.903 | 0.719 | 0.934 | 0.900 | 0.871 | 0.696 |
| RMSE | Before QA filter | MAIAC | 0.221 | 0.144 | **0.085** | **0.195** | **0.109** | **0.163** | 0.170 |
| | | DT | **0.178** | **0.131** | 0.172 | 0.275 | ------- | 0.246 | **0.097** |
| | | DB | 0.276 | 0.174 | 0.155 | 0.239 | 0.214 | 0.244 | 0.210 |
| | After QA filter | MAIAC | 0.219 | 0.127 | **0.114** | 0.194 | **0.110** | **0.161** | 0.108 |
| | | DT | **0.173** | 0.124 | 0.164 | 0.288 | ------- | 0.223 | **0.106** |
| | | DB | 0.228 | **0.122** | 0.191 | **0.159** | 0.170 | 0.177 | 0.208 |
| Bias | | MAIAC | **-0.004** | 0.066 | -0.018 | **0.030** | -0.046 | **0.002** | 0.021 |

| | | | Cropland | Forest | Grassland | Built-up | Unoccupied land | Mixed | Ocean |
|---|---|---|---|---|---|---|---|---|---|
| | Before QA filter | DT | 0.065 | **0.020** | 0.048 | 0.201 | ------- | 0.167 | **0.006** |
| | | DB | 0.092 | 0.038 | **0.011** | 0.061 | **-0.007** | 0.057 | -0.088 |
| | After QA filter | MAIAC | **-0.007** | 0.053 | 0.051 | 0.027 | **-0.057** | 0.011 | **0.031** |
| | | DT | 0.064 | **-0.003** | 0.075 | 0.224 | ------- | 0.114 | -0.057 |
| | | DB | 0.062 | -0.020 | **-0.048** | 0.019 | -0.092 | **0.007** | -0.128 |
| Within_EE | Before QA filter | MAIAC | **75.56** | 43.09 | **83.68** | **72.79** | **67.44** | **65.31** | 67.96 |
| | | DT | 71.12 | 56.23 | 47.19 | 24.36 | ------- | 38.23 | **81.11** |
| | | DB | 57.37 | **64.41** | 63.21 | 63.60 | 36.54 | 47.19 | 53.36 |
| | After QA filter | MAIAC | **76.22** | 47.75 | 46.02 | **74.66** | **71.43** | 66.80 | **78.22** |
| | | DT | 72.67 | 56.53 | 37.04 | 19.33 | ------- | 50.00 | 75.20 |
| | | DB | 60.37 | **72.20** | **60.41** | 69.24 | 37.50 | 59.34 | 51.90 |

**Table 4. Comparison of the retrieval accuracy of the MAIAC, DT and DB products for different land cover types before and after the QA filter. "-------" means no matchup pairs or that the matchup pairs number less than 10.**

Table 5 shows the validation accuracy for three products after the QA filter in four seasons. In cropland, the retrieval accuracies in autumn for the three products were better than in other seasons. For forest land types, three products showed a higher correlation in autumn than the other seasons, but the Within_EE values demonstrated the best results in winter, and the corresponding results for DB products were clearly higher than for the other two products. In terms of grassland type, MAIAC and DB products were more accurate in summer and spring, respectively. In the built-up region, all products showed a high correlation in all seasons, but DT products were seriously overestimated. In unoccupied land, matchup pairs for MAIAC and DB products were more focused in spring, and MAIAC products performed better than DB products. A high correlation was also found for the three product in mixed and ocean regions in all seasons, but more MAIAC retrievals met the EE envelope line.

| | | | Cropland | Forest | Grassland | Built-up | Unoccupied land | Mixed | Ocean |
|---|---|---|---|---|---|---|---|---|---|
| R | MAIAC | Spring | 0.912 | 0.902 | **0.956** | 0.929 | **0.945** | 0.848 | **0.951** |
| | | Summer | 0.940 | 0.856 | 0.932 | 0.951 | ------- | **0.932** | 0.917 |
| | | Autumn | **0.956** | **0.930** | 0.798 | **0.965** | ------- | 0.877 | 0.903 |
| | | Winter | 0.910 | 0.881 | 0.853 | 0.888 | ------- | 0.892 | 0.893 |
| | DT | Spring | 0.959 | 0.889 | 0.818 | 0.920 | ------- | 0.868 | 0.933 |
| | | Summer | 0.961 | 0.827 | **0.861** | 0.934 | ------- | **0.902** | ------- |
| | | Autumn | **0.983** | **0.939** | 0.608 | 0.914 | ------- | 0.834 | **0.983** |
| | | Winter | 0.939 | 0.854 | ------- | ------- | ------- | 0.869 | 0.955 |
| | DB | Spring | 0.950 | 0.917 | **0.911** | 0.954 | **0.903** | 0.828 | 0.874 |

| Metric | Product | Season | | | | | | | |
|---|---|---|---|---|---|---|---|---|---|
| | | Summer | 0.938 | 0.796 | 0.659 | **0.971** | ------- | **0.926** | ------- |
| | | Autumn | **0.943** | **0.914** | 0.714 | 0.958 | ------- | 0.856 | 0.810 |
| | | Winter | 0.931 | 0.901 | 0.617 | 0.955 | ------- | 0.874 | **0.928** |
| Bias | MAIAC | Spring | -0.021 | 0.045 | 0.065 | -0.023 | **-0.061** | -0.018 | 0.032 |
| | | Summer | 0.085 | 0.079 | 0.049 | 0.123 | **-------** | 0.073 | 0.073 |
| | | Autumn | **-0.007** | 0.065 | **0.042** | 0.054 | ------- | 0.035 | 0.044 |
| | | Winter | -0.039 | **0.034** | 0.051 | **0.013** | ------- | **-0.011** | **0.015** |
| | DT | Spring | 0.105 | **-0.006** | 0.197 | 0.226 | ------- | 0.139 | **-0.004** |
| | | Summer | 0.077 | 0.021 | 0.130 | 0.249 | ------- | 0.139 | ------- |
| | | Autumn | **0.029** | 0.008 | **0.009** | **0.194** | ------- | 0.125 | -0.088 |
| | | Winter | 0.027 | -0.027 | ------- | ------- | ------- | **0.076** | -0.073 |
| | DB | Spring | 0.040 | -0.046 | **0.025** | **-0.013** | -0.108 | **-0.021** | -0.427 |
| | | Summer | 0.061 | 0.009 | -0.027 | -0.070 | **-------** | -0.047 | ------- |
| | | Autumn | **0.029** | **-0.008** | -0.116 | 0.033 | ------- | 0.029 | -0.201 |
| | | Winter | 0.120 | -0.023 | -0.125 | 0.100 | ------- | 0.026 | **-0.038** |
| Within_EE | MAIAC | Spring | 75.31 | 50.31 | 45.43 | 76.55 | **66.67** | 64.39 | 79.86 |
| | | Summer | 68.32 | 39.85 | **64.74** | 56.83 | **-------** | 51.27 | 53.70 |
| | | Autumn | **80.83** | 45.21 | 49.12 | 74.12 | ------- | 67.38 | 76.00 |
| | | Winter | 76.58 | **53.01** | 30.56 | **83.02** | ------- | **76.87** | **83.11** |
| | DT | Spring | 65.13 | 54.81 | 6.25 | 20.47 | ------- | 45.89 | **79.41** |
| | | Summer | 63.66 | 50.00 | 20.63 | **22.49** | ------- | 40.48 | ------- |
| | | Autumn | **83.94** | 55.33 | **55.42** | 14.29 | ------- | 51.57 | 71.43 |
| | | Winter | 77.78 | **63.74** | ------- | ------- | ------- | **55.23** | 75.00 |
| | DB | Spring | 58.69 | 61.22 | **66.03** | 67.94 | **33.33** | 52.11 | 0.00 |
| | | Summer | 60.37 | 70.09 | 64.91 | 66.88 | **-------** | 54.39 | ------- |
| | | Autumn | **65.82** | 74.89 | 54.10 | **82.06** | ------- | 56.31 | 4.35 |
| | | Winter | 56.57 | **78.37** | 53.70 | 62.59 | ------- | **68.89** | 84.21 |

**Table 5. Comparison of the retrieval accuracy of the MAIAC, DT and DB products for different land cover types in four season after the QA filter. "-------" means no matchup pairs or that the matchup pairs number less than 10.**

The Ångström exponent (AE) is a key parameter to describe aerosol particle size, and in general, local aerosol sources play a dominant role in aerosol regimes (Mhawish et al., 2019). To discover aerosol particle sizes in different land covers, Figure 6 shows a scatterplot of the AE (440 nm-675 nm) parameter versus AOD for different land cover types. Our results were similar to those of Martins et al., 2017. The aerosol types in China are mainly fine-mode aerosol particles (AE>1). Some coarse-mode particles (AE<0.5) are mainly found in some regions with sparse vegetation, e.g., grassland (vegetation coverage in selected

site less than 20%), built-up and unoccupied land. As observed in Figure 3, high AOD values mainly occurred in cropland and built-up areas. According to the AE parameter, the aerosol types for these high AOD values were mainly fine-mode aerosol particles. Figure 7 presents the AOD bias distribution along with the AE parameter. A higher AOD bias often occurred when the AODs were higher than 0.8 with 1<AE<1.5. There was no AE dependence when the AOD were very small, e.g., lower than 0.1, for the three products. However, MAIAC seemed to have a more positive bias than the DB product at a very small level.

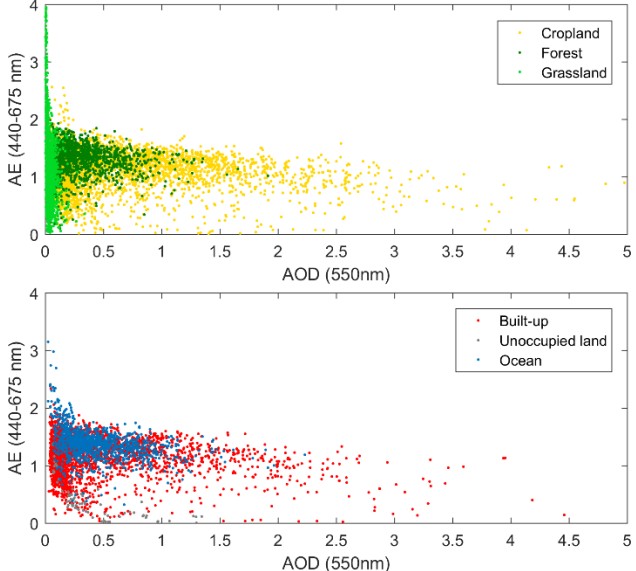

**Figure 6. Scatterplot of AOD at 550 nm against the Ångström exponent for different land cover types. We selected AERONET sites with maximum observations for each land cover type: Xianghe (cropland); Taipei_CWB (forest); QOMS_CAS (grassland); Beijing (built-up); Dunhuang (unoccupied land); Hong_Kong_PolyU (ocean).**

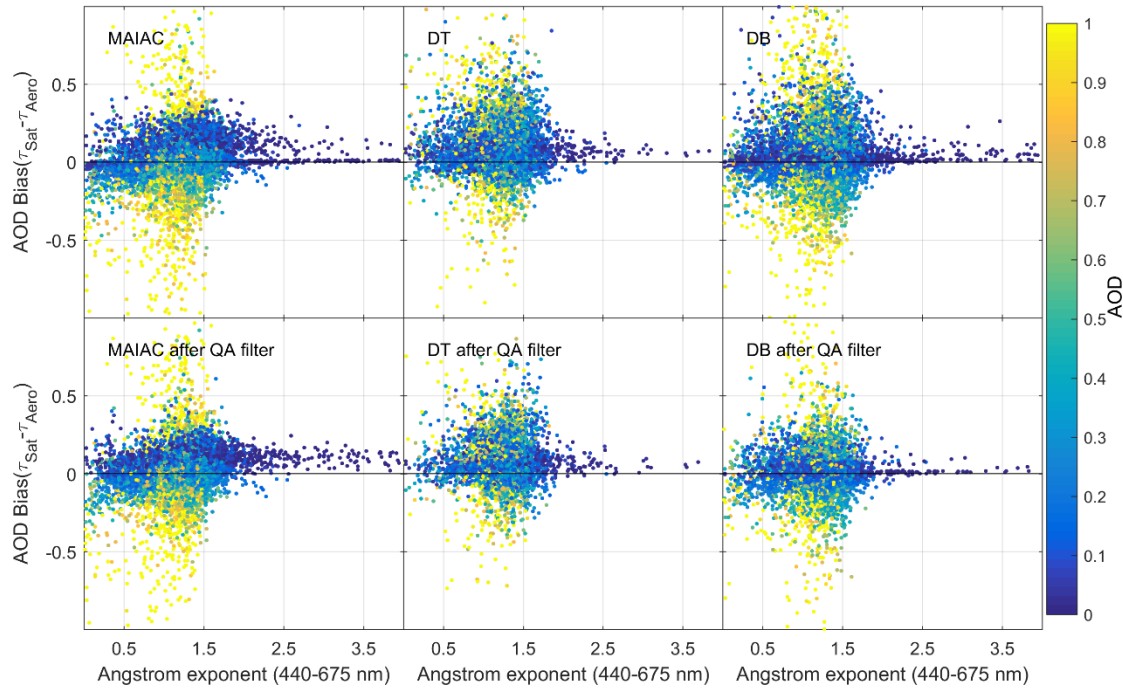

**Figure 7. Scatterplot of the AOD bias from matchup data versus the AERONET Ångström exponent (440 nm-675 nm) before and after the QA filter.**

### 4.3 View of the geometry dependency analysis

To determine how the view geometry influence the accuracy for three retrieval algorithms, we analysed view geometry dependency using the following four angles: solar zenith angle (SZA), view zenith angle (VZA), scattering angle (SA) and relative azimuth angle (RAA) (Superczynski et al., 2017). We separated each kind angle into 10 bins and statistically analysed the AOD bias distribution in each bin. The results are displayed in Figure 8.

In terms of the solar zenith angle, the three retrieval algorithms all showed a strong dependency with different characteristics. A slight downtrend along with SZA was found in the MAIAC algorithm, and the MAIAC retrievals seemed slightly overestimated when SZA was less than 40° and underestimated when it as larger. The mean biases only fluctuated between -0.05 and 0.05. For the DT algorithm, the mean bias first arose when the SZAs were small, and the mean bias reached the maximum at SZA≈25°. Then, the mean biases decreased as SZA increased. The mean biases were close to zero when SZA reached the maximum value. With regard to the DB algorithm, the mean bias first slowly decreased when the SZAs were less than 35° and then rapidly rose as SZA increased. After the QA filter, the whole mean bias line shifted down.

The MAIAC and DB algorithm showed no dependency on the view zenith angle, and the corresponding mean bias lines did not fluctuate much along with VZA. Compared with the results obtained before and after the QA filter, the mean bias line for the MAIAC algorithm slightly increased, and the mean bias line for the DB algorithm moves down to a relatively great degree. VZA slightly affected the DT performance with a little downtrend. After the QA filter, the mean bias line slightly declined.

The scattering angle also greatly impacted the performance of the three retrieval algorithms. MAIAC retrievals seemed to be underestimated when the SAs were less than 100° and slightly overestimated when they were between 100° and 155°. When the SAs were larger than 155°, the retrievals tended to be underestimated. After the QA filter, the corresponding retrievals at large SAs tended to be overestimated. For the DT and DB retrievals, a significant uptrend was observed for the mean bias along with SAs. Small positive biases were found when the SAs were very small, and large positive biases occurred when the SAs were very large. After the QA filter, the significant uptrend was alleviated for DB retrievals, but a large negative bias was found when SA approached 180°. We consider the scarce matchup number of DB retrievals to be the main reason for the negative bias.

For the MAIAC algorithm, positive biases occurred as RAA approached the extremes of 0°, 180°, and negative bias emerged as RAA neared 90°, where the matchup numbers were very limited in the three angle intervals. In the rest angle intervals, MAIAC showed no dependency on RAA. After the QA filter, a downtrend of the mean bias was apparent along with RAA during backscattering (RAA<90°), and an uptrend of the mean bias was observed during forward-scattering (RAA>90°). For the DT algorithm, the positive mean bias decreased as RAA increased upon backscattering and first increased and then decreased upon forward-scattering. After the QA filter, the downward trend tended to be alleviated upon backscattering. For the DB algorithm, upon backscattering, the positive mean bias first decreased from very high to zero and then increased to become somewhat high. Upon forward-scattering, the positive mean biases were all larger than 0.05. After the QA filter and upon backscattering, no dependency on RAA was observed for the DB algorithm, but the highest mean bias was lower than zero. Upon forward-scattering, an obvious linear downtrend from positive to negative bias was observed as RAA increased.

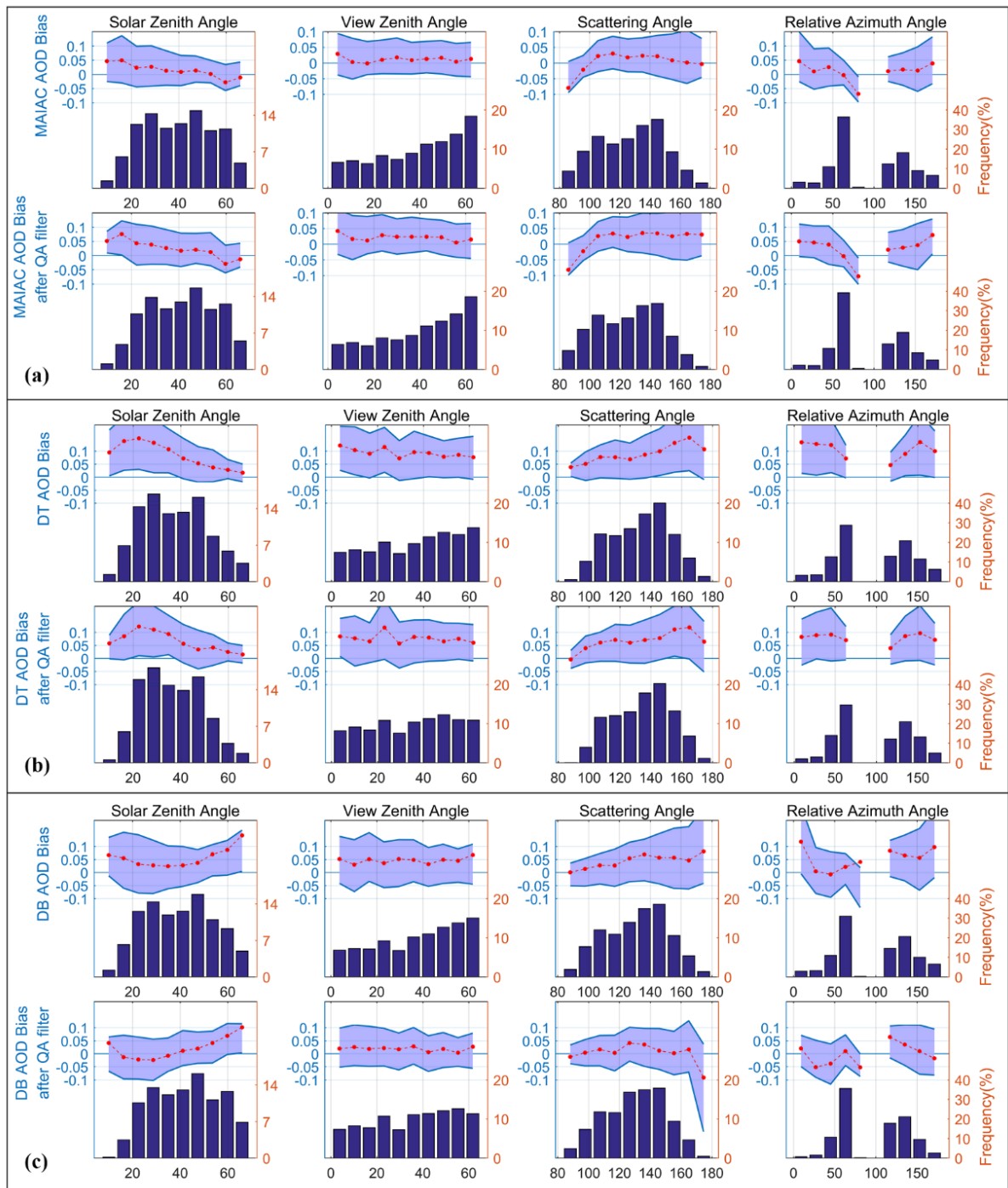

**Figure 8. Dependency of the AOD bias on the solar zenith angle, view zenith angle, scattering angle and relative azimuth angle for the (a) MAIAC product, (b) DT product and (c) DB product before and after the QA filter. The dark blue bar is the histogram bin, red points in the shadowed area are the mean bias of the corresponding bin, and the top and bottom blue line are the 75% and 25% percentiles of the AOD biases in the corresponding bin, respectively.**

## 4.4 Analysis of the spatiotemporal retrieval accuracy

To investigate retrieval accuracy of the three algorithms at different regions and different times, Figure 9 shows the R, RMSE, Bias and Within_EE results for each AERONET site, ignoring the sites with fewer than 10 matchup numbers, which might cause unreliable statistical results.

Three products presented different retrieval accuracies in different regions. In the BTH region (marked by the black box in Figure 1), three products showed a good correlation with the ground measurements, e.g., R>0.9. There were, however, more retrievals for MAIAC and DT products falling within the EE limits than the DB product. Based on the Bias results, the DT and DB products seemed to be overestimated compared with the MAIAC product. The DT product was more positively biased compared with the MAIAC product. In the YRD region, the within_EE results showed that more MAIAC retrievals met the

EE limits than DT and DB products. A good correlation of the three products was also found in this region. However, the DT product was overestimated, and DB was underestimated. In the PRD region, the MAIAC retrievals were obviously more accurate than the DT and DB retrievals. The Within_EE results for the MAIAC retrievals in this region were all greater than 70%. The Within_EE results for the DT retrievals were relatively low for some sites before the QA filter. After the QA filter, the Within_EE results were greatly promoted. DB retrievals in this region demonstrated the worst performance with low

Within_EE results, a bad correlation and a negative bias. In addition, the MAIAC product was also the most accurate product in the NW area. The Within_EE and R results overall were higher than for the DT and DB products. Additionally, the RMSE results for the MAIAC product in this region were also relatively lower than those for the BTH and YRD region. The Within_EE results for the MAIAC product for most sites in the west of Taiwan were higher than 66% after the QA filter, demonstrating a high accuracy compared with DT and DB products. However, according to the east site, e.g., Lulin, the

MAIAC retrievals seemed to be overestimated with low Within_EE and R results. Additionally, DB retrievals in the Lulin site seemed to be unbiased with high Within_EE (over 70%) and relatively high R (over 0.8) results. In the Tibet area, three algorithms all failed to retrieve AODs according to the statistical results due to the high latitude and snow cover.

Figure 10 presents the monthly validation results for the three products. We overlooked the specific QOMS_CAS site for this purpose due to its poor performance after the QA filter, which would affect the overall accuracy. Three products showed a

good correlation with the ground measurements for all months with R>0.85. The AOD deviation for the DT and MAIAC products was higher in spring and summer than autumn and winter, consistent with the results of He et al., 2017 (He et al., 2017). The RMSE results for the DB products were generally higher than the DT and MAIAC products before the QA filter. After the QA filter, the RMSE results decreased with no obvious seasonal variability law. The DT product seemed to be systematically overestimated, and the positive biases were extremely high in spring and summer. The MAIAC product was

positively biased from June to October with a Bias<0.1. The DB product was positively biased in all seasons before the QA filter, but the Bias results from June to October were significantly reduced after the QA filter. Before the QA filter, the Within_EE results for the MAIAC product were higher than the DT and DB products in all months. However, less than 60% of the MAIAC retrievals fell within the EE limits in summer. After the QA filter, the Within_EE results for the DB product

from June to September were superior to those of the MAIAC and DT products. The $R^2$ results for the MAIAC products were stable for all months, and most $R^2$ results were over 0.8. The DB product had a lower $R^2$ in the cold season from November to February, and in April and May, the $R^2$ results for the DT product were generally lower than those in the other months. After the QA filter, the DB product achieved higher $R^2$ results from April to September. According to the matchup number results, the MAIAC product had more matchup numbers than the DT and DB products. However, all products had fewer matchup numbers in summer due to the increased cloud cover in the rainy season. In summary, the MAIAC product was more accurate than the DT and DB products expect for in summer season. In contrast to the positive bias of MAIAC retrievals in summer, the DB product after the QA filter could achieve unbiased results with higher Within_EE and $R^2$ than the MAIAC product.

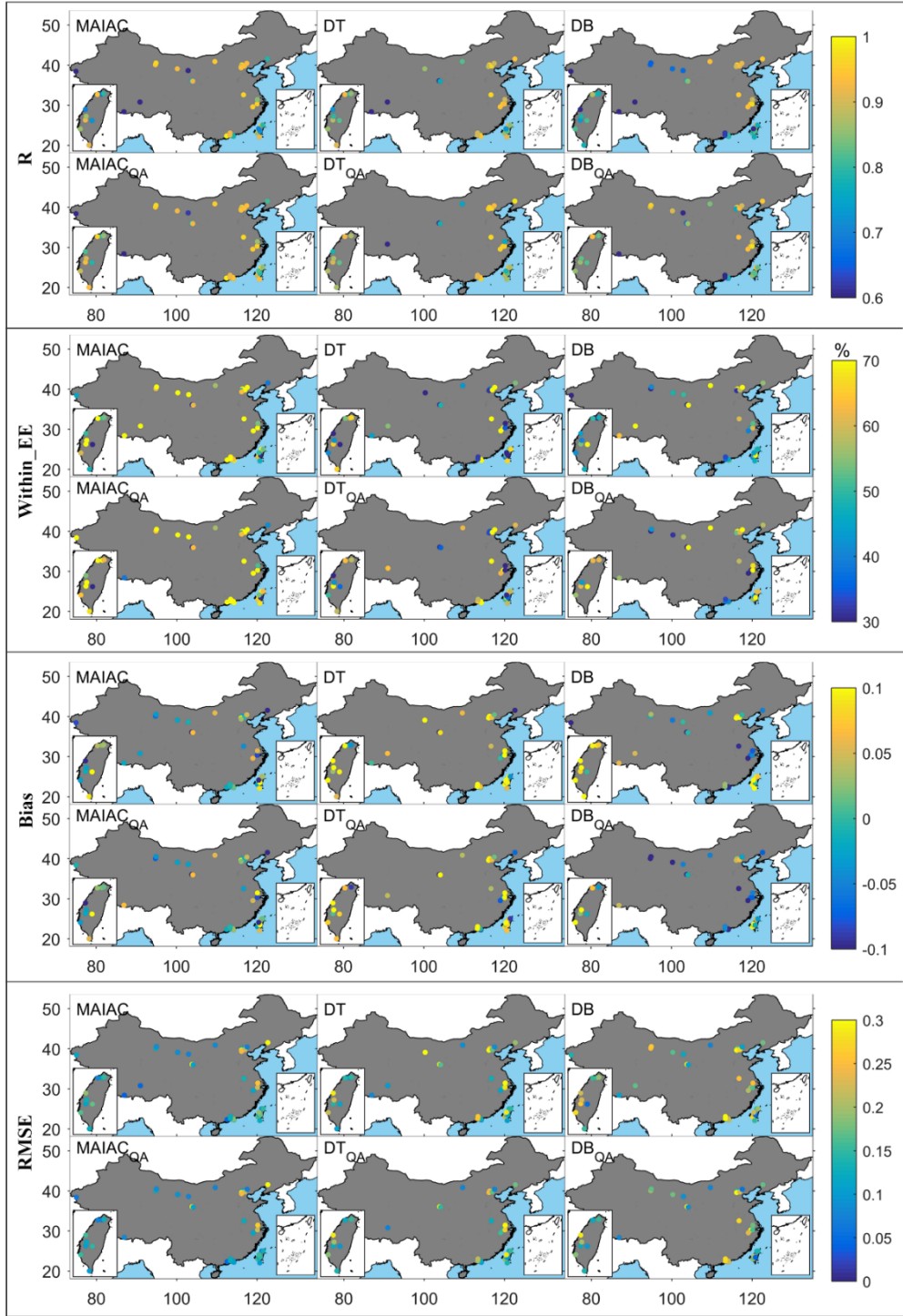

**Figure 9. Evaluation results for MAIAC, DT, and DB after and before the QA filter in each AERONET site. The subscript QA denotes the corresponding results after the QA filter.**

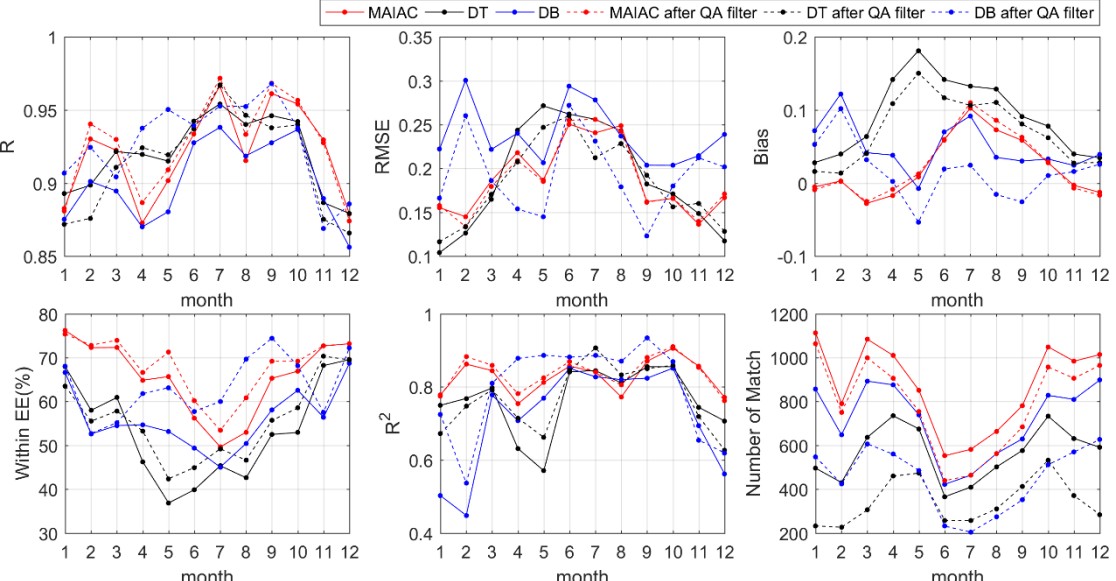

**Figure 10. Validation of MAIAC, DT and DB in different months before and after the QA filter.**

To investigate the annual change in retrieval accuracy for three products to ascertain whether MODIS instrument maintain its performance due to it exceeding its designed lifetime. However, according to Table 1, the time durations of each AERONET

site were significantly different. Thus, the matchup observation pair during each year were from different sites. This phenomenon may result from incomparable validation results for each year. However, if only considering the sites with the same monitoring time, most sites will be discarded, and fewer matchup numbers will cause unreliable corresponding statistical results. Thus, we still adopted all site measurements. We ignored the results for the years 2000, 2001, 2002 and 2003 due to fewer matchup numbers in these years. According to Figure 11, three products showed a high correlation with ground

measurements according to the R and $R^2$ results, except in 2009. The reason for the sharp decline in the R and $R^2$ results in 2009 was mainly that some sites, e.g., Qiandaohu, SACOL, Kaiping, Shouxian, and Zhangye, did not have matchup pairs in this year, and matchup pairs containing bad retrieval satellite pixels around the Lanzhou_City and NAM_CO sites appeared in 2009. Thus, the high correlation revealed MODIS instrument results consistent the AOD retrievals from 2000 to 2017. The MAIAC AOD deviation was generally small, with most RMSE results being lower than 0.2 and larger than 0.15. The RMSE

results for the DB product were generally larger than 0.2 before the QA filter. After the QA filter, the RMSE results varied in a large range from 0.15 to 0.25. Based on the Bias result, there was a significant uptrend for the three products over the year. The MAIAC Bias results were generally smaller than the DT and DB products, and most Bias results for the MAIAC product were within ±0.05, with a negative bias before 2010 and a positive bias after 2010. To eliminate the influence of the contribution of some specific sites in the specific year, Figure 12 plots bias time series for five AERONET sites with a

monitoring time covering most study years and ignoring data with matchup numbers less than 20. The bias uptrend seemed to appear in three products for all selected sites except the EPA-NCU site for DT products. Thus the significant uptrend of Bias results is not caused by the significantly different time durations of AERONET sites. For the Within_EE results, MAIAC also

showed better accuracy than DT and DB products, and for Within_EE results, a slight declining trend was observed over the year. The matchup numbers for the three products revealed an increasing trend due to the establishment of greater numbers of AERONET sites in the China region over time.

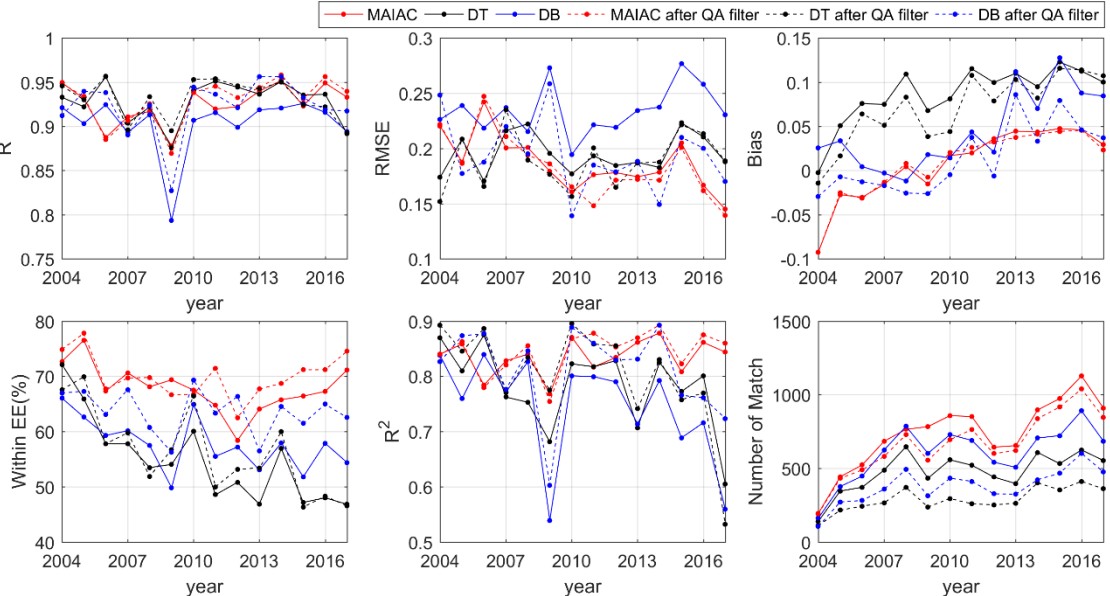

5 **Figure 11. Validation of MAIAC, DT and DB in different years before and after the QA filter from 2004 to 2017.**

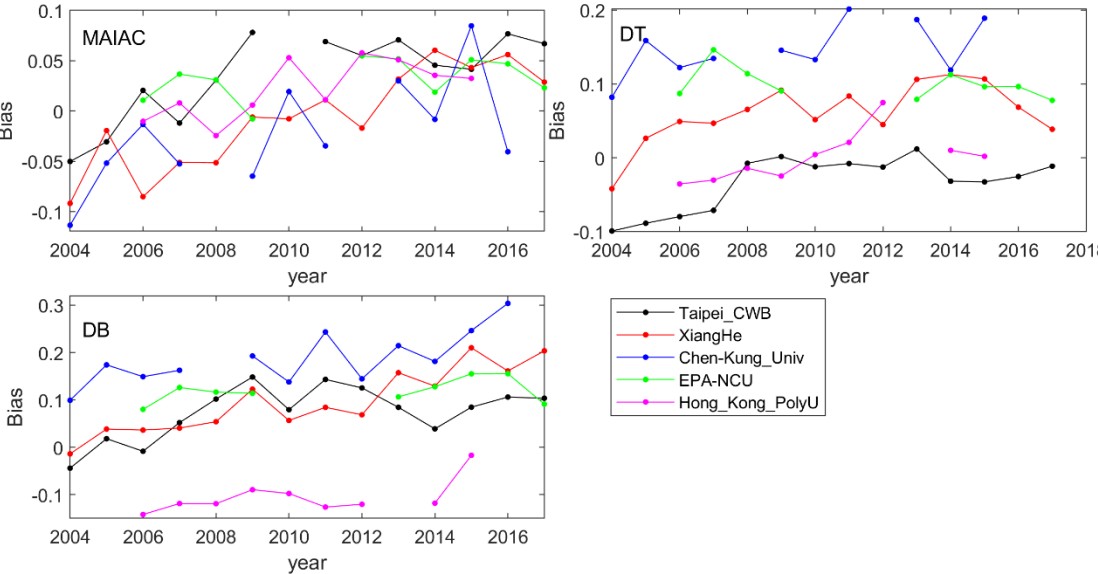

**Figure 12. Bias plot for the three products before the QA filter in five selected AEROENT sites with a monitoring period containing most of the study years from 2004 to 2017.**

## 4.5 Analysis on spatial pattern variation difference

To compare the difference in spatial variations for the three products, we upscaled the MAIAC product to match the grid of the DT and DB products; thus, 1 km pixels falling within the 10 km grid were averaged. Such a protocol can aid in investigating differences in different regions between the three products.

Figure 13 presents multiyear averaged and difference results between MAIAC, DT and DB products, with aerosol loading presenting a noteworthy assembly characteristic. Higher AOD values were concentrated in the North China Plain and Szechwan Basin where the land cover types were mainly cropland-oriented, as shown in Figure 1. Before the QA filter, compared with the DT and DB observations, the MAIAC AODs were smaller in the North China Plain and larger in Yunnan Province and East Taiwan. After the QA filter, the DB AODs became smaller in the North China Plain and southeast region.

Compared with the DB AODs, the MAIAC AODs became slightly higher in the North China Plain (difference over 0.1) and obviously higher in Southeast China (difference over 0.3). Recall the statistical result presented in Figure 9, in which the DT and DB products were overestimated in the BTH region, the DB product was underestimated in the YRD region, and the MAIAC product seemed to be overestimated in East Taiwan. These findings indicate that MAIAC retrievals are more accurate than DT and DB in the North China Plain and southeast region, and DB retrievals are more accurate than MAIAC in East

Taiwan. However, due to the lack of the AERONET site in Yunnan Province, we could not evaluate the accuracy of the three products in Yunnan Province. The difference before and after the QA filter for the MAIAC product was very small, except for some individual pixels in the Tibet region. In addition, there was an obvious boundary in the 30° latitude for MAIAC AODs. This boundary was caused by the different regional aerosol model used above and below 30° latitude (Lyapustin et al., 2018).

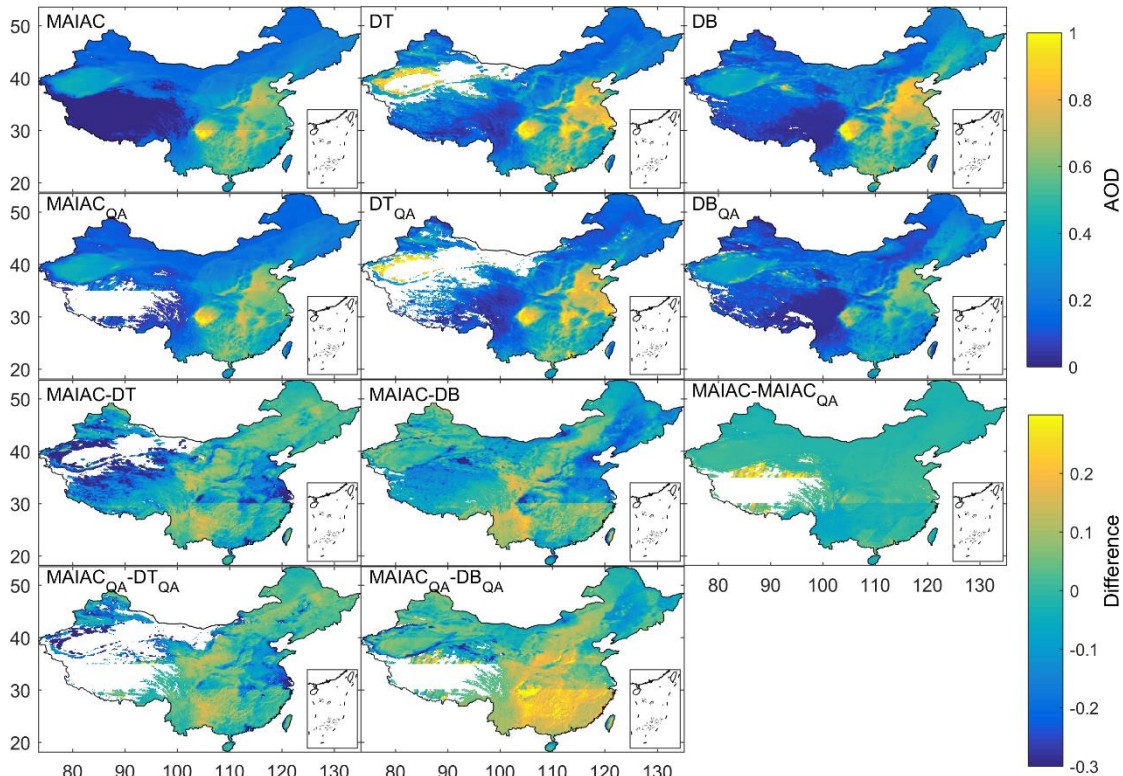

**Figure 13. Averaged AOD distributions throughout the year for MAIAC, DT, and DB before the QA filter and their differences after the QA filter from 2000 to 2017. The subscript QA denotes the corresponding results after the QA filter.**

Figure 14 show the seasonal comparison results among three products before and after the QA filter. The AOD spatial variation
5   for the three products showed apparent seasonal characteristics. The AODs in the North China Plain in summer were higher
than in other seasons, and the AODs in the Tarim Basin in spring were higher than in other seasons. Based on the AOD spatial
variation difference map, the difference between MAIAC and DT in the North China Plain evolved gradually from negative
in spring to positive in winter. The negative difference between MAIAC and DB in the North China Plain was higher in
summer and winter than in spring and autumn. The positive difference in Yunnan Province between MAIAC and DT was
10   slightly lower than that between MAIAC and DB. After the QA filter, AODs in South China for the DB product were extremely
low compared with those for the MAIAC product.

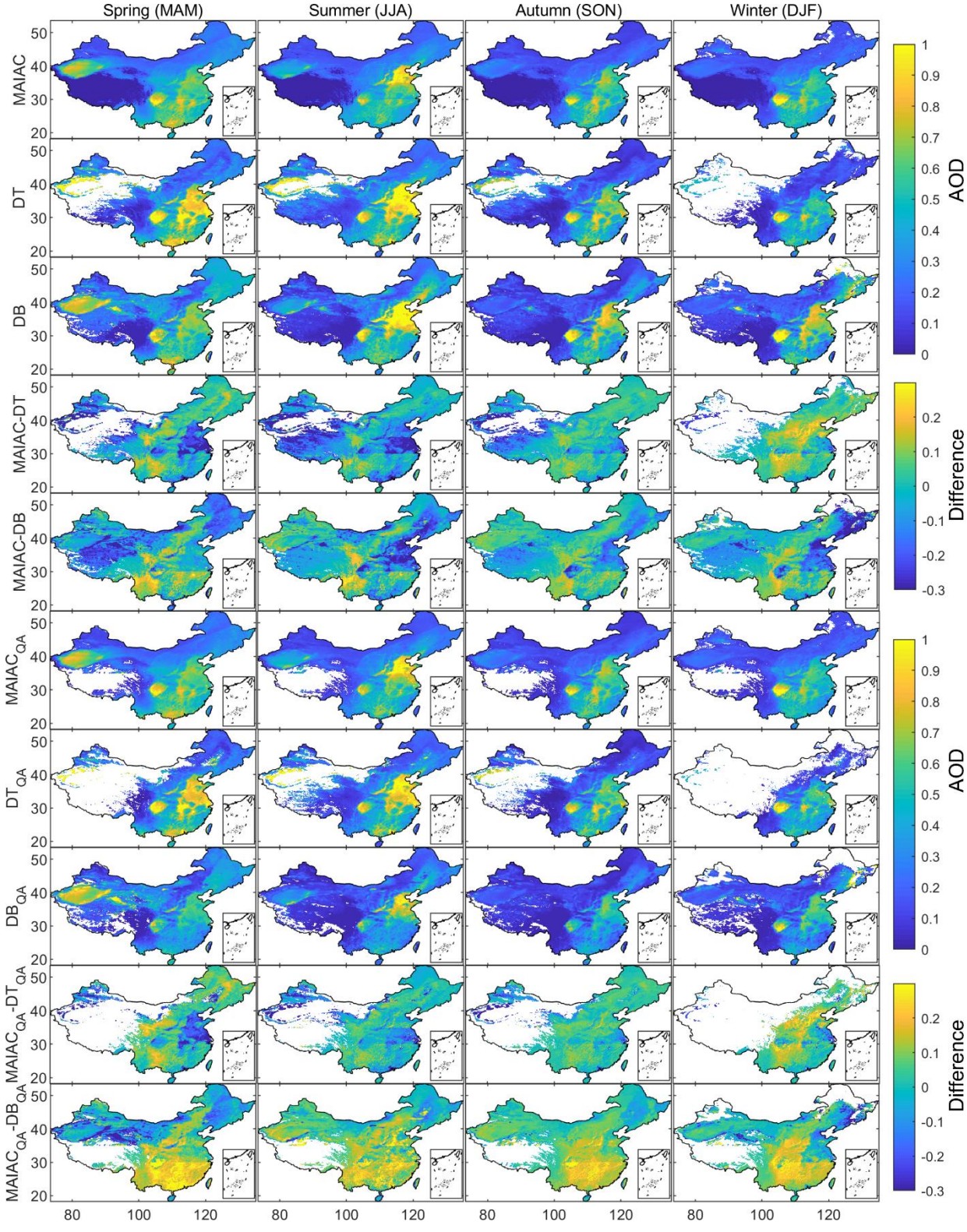

**Figure 14. Seasonal averaged AOD distributions for MAIAC, DT, and DB and their differences before and after the QA filter from 2000 to 2017. The subscript QA denotes the corresponding results after the QA filter.**

### 4.6 Analysis of spatiotemporal completeness

Based the upscale MAIAC 10 km data in section 4.4, the spatial completeness in equation (7) and temporal completeness in equation (8) for three products are showed in Figure 15 and Figure 16. According to Figure 15, the spatial completeness of the MAIAC product was higher than the DT and DB products before and after the QA filter. The spatial completeness of the DT product was smallest due to its retrieval failure on a bright surface. The spatial completeness for all the products showed an obvious periodical trend change. Table 6 shows the statistics for the spatial completeness of the three products in different seasons. Before the QA filter, the averaged spatial completeness of MAIAC (46.87%) was higher than DT (16.66%) and DB (34.80%). After the QA filter, the reduced proportion of MAIAC (17.18%) exceeded DB (15.30%) and DT (8.66%) because many climatology values in the Tibet Plateau were discarded. Comparison of the spatial completeness in four seasons revealed a higher spatial completeness for the three products in autumn than the other three seasons due to the reduced cloudiness in the dry autumn season. The spatial completeness in winter was smallest due to the influence of the surface snow cover and large deciduous trees. Compared with MAIAC and DB products, the spatial completeness of the DT product in winter was minimal due to the bright surface in winter.

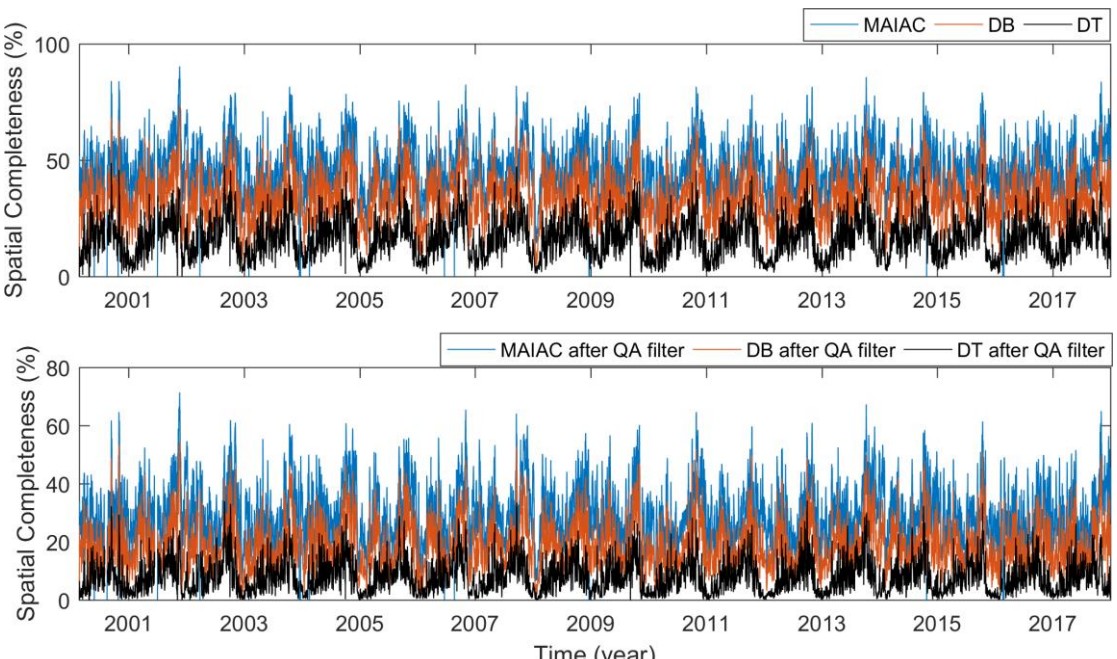

**Figure 15. Daily spatial completeness for MAIAC, DT and DB from 2000 to 2017 before and after the QA filter.**

|  |  | All year | Spring | Summer | Autumn | Winter |
|---|---|---|---|---|---|---|
| Before | MAIAC | 46.87 | 44.80 | 43.83 | 55.80 | 42.89 |

| | | | | | | |
|---|---|---|---|---|---|---|
| QA filter | DT | 16.66 | 15.71 | 19.22 | 22.72 | 8.60 |
| | DB | 34.80 | 34.93 | 33.59 | 42.01 | 28.30 |
| After QA filter | MAIAC | 29.69 | 29.06 | 25.17 | 37.17 | 27.22 |
| | DT | 8.00 | 7.20 | 9.63 | 11.76 | 3.19 |
| | DB | 19.50 | 20.31 | 16.23 | 25.90 | 15.31 |
| Declined Proportion | MAIAC | 17.18 | 15.74 | 18.66 | 18.63 | 15.67 |
| | DT | 8.66 | 8.52 | 9.59 | 10.96 | 5.40 |
| | DB | 15.30 | 14.61 | 17.36 | 16.12 | 12.99 |

**Table 6. Seasonal averaged spatial completeness for MAIAC, DT, DB before and after the QA filter and their declining proportions after the QA filter**

Figure 16 presents the temporal completeness in China for the three products. Due to the climatology values in the Tibet Plateau, the temporal completeness of the MAIAC product in this region was very high (over 80%). After the QA filter, the temporal completeness rapidly decreased in this region. In the other region, the declining proportions of temporal completeness for MAIAC were mostly lower than 10%, except for Yunnan Province (nearly 15%), Hainan Province (nearly 20%) and East Taiwan (nearly 20%). Compared with the MAIAC and DB products, DT retrievals were very scarce in Tarim Basin due to failure on the bright desert surface. DT retrievals were more concentrated on the North China Plain and Yunnan Province. After the QA filter, a dramatically reduced proportional area of temporal completeness (nearly 30%) for DT products was observed in the cropland region in Northeast China. The severely reduced proportional area (nearly 40%) for the DB product after the QA filter was mainly focused on unoccupied land, e.g., gobi, saline-alkali soil, etc., at the top of the Tibet Plateau. Compared with the MAIAC product, before the QA filter, the DB product showed more retrievals in the Tarim Basin, North China Plain and Southeast China, and fewer retrievals in Yunnan Province and Northeast China. After the QA filter, the temporal completeness of the MAIAC product was better than the DB product in all regions.

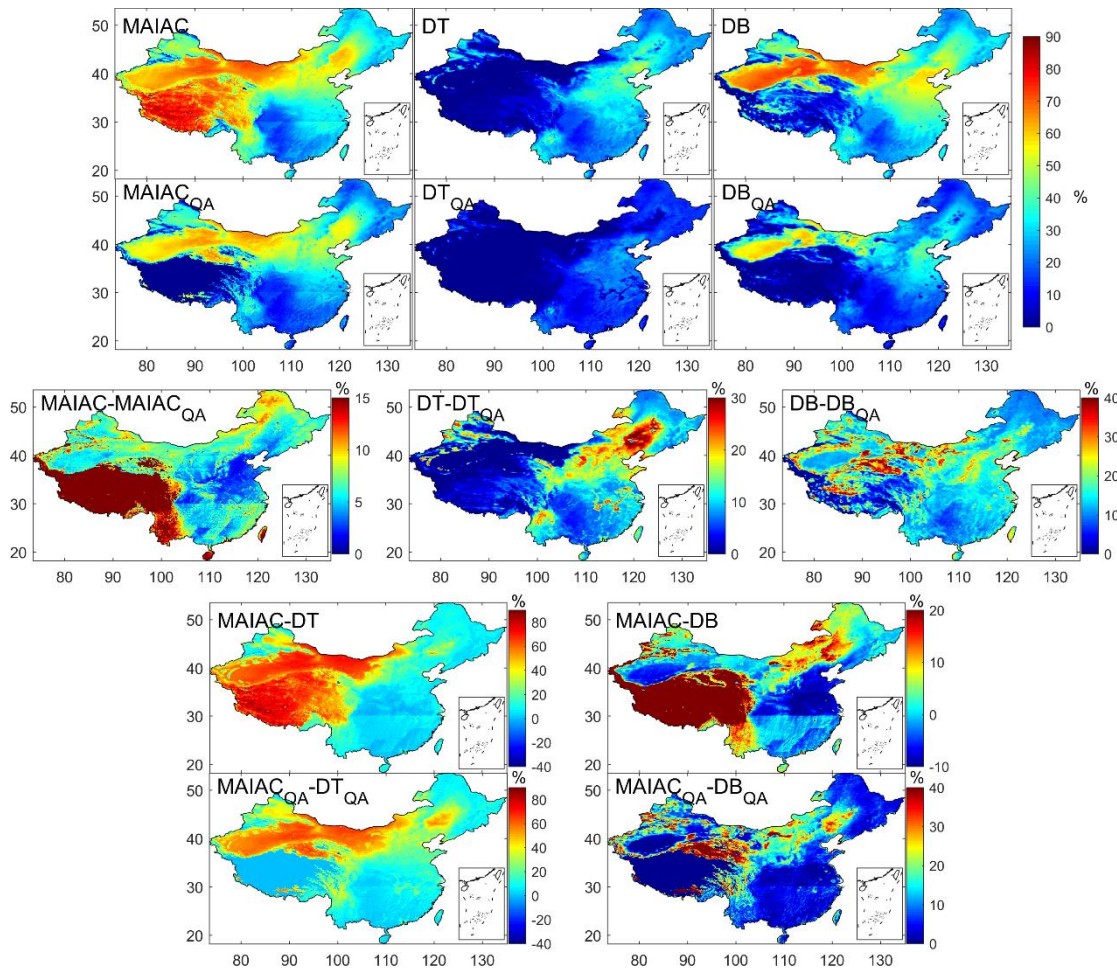

**Figure 16. Spatial distributions of temporal completeness for MAIAC, DT, and DB before and after the QA filter and their differences from 2000 to 2017. The subscript QA denotes the corresponding results after the QA filter.**

## 5. Conclusion

5    In this study, we present the first comprehensive validation and comparison of three MODIS aerosol retrieval algorithms (i.e., MAIAC, DT and DB) across China in terms of overall accuracy, land cover dependency, viewed geometry dependency, spatiotemporal retrieval accuracy, spatial distribution difference and spatiotemporal completeness. These validation results may guide users to utilize the three products appropriately. The main results and conclusion are presented below.

−    In terms of overall accuracy, the MAIAC product is more accurate than the DT and DB products. The DT and DB products
10      are positively biased before the QA filter, and the positive bias for the DB product is alleviated by the QA filter.

−    DT retrievals in cropland, forest and ocean seem to be more accurate but with a positive bias than retrievals by the MAIAC and DB algorithms. The MAIAC algorithm performs better in grassland, built-up and mixed areas than the DT and DB algorithms.

- Three algorithms show a strong dependency on SZA, SA and RAA. VZA only marginally affects the retrieval accuracy of the three algorithms.

- The MAIAC product performs better in the BTH, YRD, PRD and NW regions than the DT and DB algorithm, and the DB product performs better than the DT and MAIAC products after the QA filter in East Taiwan. The MAIAC algorithm performs better than the DT and DB algorithms in most months except June, July, August and September. In these four months, MAIAC retrievals appear to be overestimated, and DB retrievals after the QA filter are more accurate than MAIAC retrievals.

- Three AOD products present a similar spatial pattern with high aerosol loading in the North China Plain and Szechwan Basin. In comparison, MAIAC retrievals are lower in the North China Plain and Szechwan Basin than DT and DB retrievals and are higher in Yunnan Province and East Taiwan than DT and DB retrievals. After the QA filter, the DB AOD values are significantly reduced and obviously lower than the MAIAC product in Southeast China.

- Based on spatiotemporal completeness analysis, the MAIAC product has more retrievals in the spatiotemporal domain than the DT and DB products. The spatial completeness exhibits a strong periodical change, and the temporal completeness is highest in autumn than other seasons due to the decreasing cloud cover in this dry season, which is lowest in winter due to the snow cover and deciduous vegetation. In terms of temporal completeness, MAIAC has more retrievals in the Tarim Basin and the cropland in Northeast China compared with the DT algorithm. Compared with the DB algorithm, MAIAC has fewer retrievals in the Tarim Basin and Southeast China and more retrievals in Northeast China. After the QA filter, the temporal completeness of MAIAC in all regions of China is better than the DB product.

**Author contribution**

Ning Liu and Bin Zou designed the whole experiment. Ning Liu and Yu Liang developed the experiment code and performed it. The manuscript was initially written by Ning Liu and fully revised by Bin Zou. Huihui Feng, Wei Wang and Yuqi Tang provided a lot of constructive comments on the experiment.

**Acknowledgements**

We thank the support of the National Key Research and Development Program of China (grant 2016YFC0206205), the National Natural Science Foundation of China (grant 41871317) and the Innovation Driven Program of Central South University (No. 2018CX016). We also thank the NASA for providing MAIAC, DT and DB products. We would like to thank the Yujie Wang and Alexei Lyapustin in NASA for answering my question on MAIAC product and this help me to push on my work.

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
