# Peer review of "Evaluation and comparison of MAIAC, DT and DB aerosol products over China"

_Atmospheric Chemistry and Physics, 2018_

## Referee Comment (RC1) · Anonymous Referee #1 · 15 Feb 2019

General comments: Validation of the aerosol products derived from the satellite observation is an important issue. This study gives a compressive assessment for the AOD products based three aerosol retrieval algorithms in MODIS sensor using ground-truth measurements from Aerosol Robotic Network (AERONET) sites over China. This manuscript is logically organized, the analysis methods are technically sound but not novel, and the results are interesting albeit some points not adequately illustrated. I have some comments on interpretation of the major results. As such, I recommend its publication pending the following concerns satisfactorily addressed.

Specific comments: 1. Page 3, Line5-10: the description of 'shortwave infrared band, e.g. 212 nm' is wrong. The authors mistook the unit. 2. Page 4, Line26-28: how to get the AOD at 550 nm using Ångström exponent in the two neighboring bands at

500 nm and 675 nm, which should be shown. It is a key to confirm the reliability of AERONET data as a reference to evaluate the MODIS products. 3. Page 4, line 25-31, these are about the AERONET data introduction, what's more, these are about why you choose the AERONET data as a reference to evaluate the MODIS products, which are omitted, including the reliability of AERONET measurements in China (e.g. Liu et al., Aerosol optical properties and radiative effect determined from sky-radiometer over Loess Plateau of Northwest China. 2011, ACP; Bi et al., Dust aerosol characteristics and shortwave radiative impact at a Gobi Desert of Northwest China during the spring of 2012. 2014, J. Meteo. Soc. Jp; Che et al., Column aerosol optical properties and aerosol radiative forcing during a serious haze-fog month over North China Plain in 2013 based on ground-based sunphotometer measurements. 2014, ACP). 4. The authors introduced the statistical approach, however, what's meaning of 'QA filter'? furthermore, what's meanings of 'before QA filter' and 'after QA filter'? What's the relation between the statistical approach with QA filter? These should be added in statistical approach. 5. According to the information of AERONET sites, as listed in Table 2, the time durations of data are significantly different among the sites, and the MODIS products are from 2000-2017. So, the problem is how to exclude the limitation of different temporal scales? Additionally, a table on the summary of comparison samples at each AERONET station with three MODIS products is needed. 6. In Figure 3-5, the seasonal variation of the land cover has not been considered, the land type is determined one type in entire year, as listed Table 2. However, the land cover varies in different seasons. Thus, there may be inaccuracy to evaluate the products under different land type. So, I suggest you can consider the seasonal land type in monthly and seasonal evaluation of MODIS products. 7. Figure 12 shows that the QA filter indicates little influence on the MAIAC product itself, similar to DT product. Therefore, what's the importance or role of the QA filter? 8. I suggest the authors can combine Figure 13 and 14 into one graph. 9. The time period should be described for Figure 12-14. 10. In the abstract and conclusion, the authors need tell us clearly which product is better to use under which kind of land cover type instead of specifics of bias, correlation coefficient

and so on. I suggest the authors rephrase the abstract more general.

---

## Referee Comment (RC2) · Anonymous Referee #2 · 19 Mar 2019

This manuscript is well constructed and easily followed. I found several absences of "the" and "a" and it is better to use "AEROENT stations" to replace "AEROENT sites". I suggest the authors asking a native English speaker with a geography Ph.D. to thoroughly check the English.

---

## Author Comment (AC1) · 2 Apr 2019

We would like to take this opportunity to thank the editor and the Reviewer 1 for their positive review and constructive suggestions. We have revised the manuscript based on the suggestions. All the correction in this revision process are marked by red color. Given below is a summary of the responses and revisions.

Reviewer 1: General comments: Validation of the aerosol products derived from the satellite observation is an important issue. This study gives a compressive assessment for the AOD products based three aerosol retrieval algorithms in MODIS sensor using ground-truth measurements from Aerosol Robotic Network (AERONET) sites over China. This manuscript is logically organized, the analysis methods are technically

[Figure]

sound but not novel, and the results are interesting albeit some points not adequately illustrated. I have some comments on interpretation of the major results. As such, I recommend its publication pending the following concerns satisfactorily addressed.

Specific comments: 1. Page 3, Line5-10: the description of 'shortwave infrared band, e.g. 212 nm' is wrong. The authors mistook the unit.

Response: Thanks for your carefully review on our manuscript. We have corrected the all the wrong description '212 nm' into '2119 nm' in the revised manuscript (e.g. page 3, line 12; page 4, line 18).

2. Page 4, Line26-28: how to get the AOD at 550 nm using Ångström exponent in the two neighboring bands at 500 nm and 675 nm, which should be shown. It is a key to confirm the reliability of AERONET data as a reference to evaluate the MODIS products.

Response: We have added the corresponding Ångström exponent formula in the page 5, line 10-13 to show how we calculate the AOD value at 550nm using measurements from two neighboring bands at 500 nm and 675 nm in the revised version.

3. Page 4, line 25-31, these are about the AERONET data introduction, what's more, these are about why you choose the AERONET data as a reference to evaluate the MODIS products, which are omitted, including the reliability of AERONET measurements in China (e.g. Liu et al., Aerosol optical properties and radiative effect determined from sky-radiometer over Loess Plateau of Northwest China. 2011, ACP; Bi et al., Dust aerosol characteristics and shortwave radiative impact at a Gobi Desert of Northwest China during the spring of 2012. 2014, J. Meteo. Soc. Jp; Che et al., Column aerosol optical properties and aerosol radiative forcing during a serious haze-fog month over North China Plain in 2013 based on ground-based sunphotometer measurements. 2014, ACP).

Response: Thanks for your suggestion on this point. Sentences addressing the reliability of AERONET measurements and relative references are added as suggested. And this can be found in the page 4, line 32 to page 5, line 1-5.

4. The authors introduced the statistical approach, however, what's meaning of 'QA filter'? furthermore, what's meanings of 'before QA filter' and 'after QA filter'? What's the relation between the statistical approach with QA filter? These should be added in statistical approach.

Response: We are sorry for the ambiguous description on the QA filter. Actually, snow, cloud, land cover type will increase the retrieval uncertainty of satellite based AOD. In order to help users to select the satellite based AODs with best quality, DT/DB/MAIAC AOD products provide a QA flag to indicate their retrieval uncertainties. QA=3 means good quality for DT algorithm, QA=2,3 means good quality for DB algorithm, and the 8 11 byte (bits "0000") of "AOD QA" SDS datasets in MAIAC products means good quality. In this study, we evaluate the accuracy improvement and the spatiotemporal completeness reduction after QA fiter. Descriptions about QA fiter were added in page 3, line 3-4,23-24; page 4, line 22-24; page 4, line 8; page 4, line 26. And we also stated "All the statistical indicators are calculated for three products before and after QA filter" in the end of section 3.3 of statistical approach.

5. According to the information of AERONET sites, as listed in Table 2, the time durations of data are significantly different among the sites, and the MODIS products are from 2000-2017. So, the problem is how to exclude the limitation of different temporal scales? Additionally, a table on the summary of comparison samples at each AERONET station with three MODIS products is needed.

Response: As pointed, the time durations of AERONET sites are significantly different. However, after adding the number of matchup pairs for three aerosol products in the Table 1. We found the distribution in each site for three aerosol products are very similar, so the matchup pairs for three aerosol products in the same AERONET site are from the same period. Thus, the validation results are still comparable between three

aerosol products. Meanwhile, it has to be acknowledged that the biggest influence caused by different time scales may be the yearly validation results in Figure 11 as the validation results in each year are calculated from different AERONET sites. However, if we only adopt the data from sites which the monitoring period cover the whole study time (i.e. 2000-2017), the eligible AERONET sites would be very less. So in this validation process, we still adopt all sites' data as previous convention. But in this process, we carefully checked any singular results presented in Figure 11 to judge whether the singular result are caused by different time scales of AERONET site, three retrieval algorithms or the MODIS sensors from Terra. All the analysis mentioned above are added in the page 27, line 3-8, line 10-13, line 18-22.

6. In Figure 3-5, the seasonal variation of the land cover has not been considered, the land type is determined one type in entire year, as listed Table 2. However, the land cover varies in different seasons. Thus, there may be inaccuracy to evaluate the products under different land type. So, I suggest you can consider the seasonal land type in monthly and seasonal evaluation of MODIS products.

Response: The validation in seasonal variation of land cover type is needed, however, the lacking seasonal land cover data is the biggest problem to do this. What's more, most common used land cover products are most yearly scale due to the less changes in short time period, including land cover data used in this study, MODIS land cover products (MCD12), etc. Instead, we consider to evaluate the seasonal performance of three satellite aerosol retrieval algorithms under the same land cover type. The corresponding results are shown in Table 5. And the analysis on Table 5 are presented in page 18, line 3-11.

7. Figure 12 shows that the QA filter indicates little influence on the MAIAC product itself, similar to DT product. Therefore, what's the importance or role of the QA filter?

Response: Although the QA filter has a little influence in the spatial pattern of averaged AOD during 2000 and 2017, the accuracy of three MODIS products after QA filter is all

improved based on the previous validated results. Thus, the QA flag is still needed to tell users to select AOD products with the best quality.

8. I suggest the authors can combine Figure 13 and 14 into one graph.

Response: Corrected.

9. The time period should be described for Figure 12-14.

Response: Corrected.

10. In the abstract and conclusion, the authors need tell us clearly which product is better to use under which kind of land cover type instead of specifics of bias, correlation coefficient and so on. I suggest the authors rephrase the abstract more general.

Response: Thanks for your suggestion, we have modified the abstract and conclusion. In these two sections, we purified major conclusion from overall validation, land cover type dependency analysis, view geometry dependency analysis, spatiotemporal retrieval accuracy analysis, spatial pattern variation difference analysis and spatiotemporal completeness analysis.

Attachment is the revised manuscript.

Please also note the supplement to this comment:
https://www.atmos-chem-phys-discuss.net/acp-2018-1339/acp-2018-1339-AC1-supplement.pdf

**Supplement:**

[revised manuscript text omitted]

---

## Author Comment (AC2) · 2 Apr 2019

We would like to thank the Reviewer 2 for your careful and positive review on the manuscript. All the correction in this revision process are marked by red color. Following is our response.

Reviewer 2

This manuscript is well constructed and easily followed. I found several absences of "the" and "a" and it is better to use "AEROENT stations" to replace "AEROENT sites". I suggest the authors asking a native English speaker with a geography Ph.D. to thoroughly check the English.

Response: We have corrected the absences of "the" and "a" and replace all

"AEROENT stations" to replace "AEROENT sites". Finally, we ask the help of a professional English editing company, e.g. American Journal Experts, to polish the whole English writing.

Please also note the supplement to this comment:
https://www.atmos-chem-phys-discuss.net/acp-2018-1339/acp-2018-1339-AC2-supplement.pdf

**Supplement:**

[revised manuscript text omitted]

---

## Author Response (AR2)

**Author's Response**

**No.**:   acp-2018-1339        Submitted on 22 Dec 2018
**Title**: Evaluation and comparison of MAIAC, DT and DB aerosol products over China
**Authors**: Ning Liu, Bin Zou, Huihui Feng, Wei Wang, Yuqi Tang, Yu Liang

We would like to thank the editor and the reviewers for their positive review on the manuscript acp-2018-1339. All the correction in this revision process are marked by red color. Following is our responses.

**Reviewer# 1:**
Please inter-compare the MODIS observation with some results of active remote sensing or model simulation and add some other remote sounding results in China.
1.   Huang, J., Minnis, P., Yi, Y., Tang, Q., Wang, X., Hu, Y., Liu, Z., Ayers, K., Trepte, C., and Winker, D., 2007. Summer dust aerosols detected from CALIPSO over the Tibetan Plateau. Geophys. Res. Lett. 34(18), L18805. doi:10.1029/2007gl029938.
2.   Jia R., Y. Liu, B. Chen, Z. Zhang, J. Huang, 2015: Source and transportation of summer dust over the Tibetan Plateau. Atmospheric Environment, 123(2015), 210–219, doi:10.1016/j.atmosenv.2015.10.038.
3.   Liu Y., Y. Sato, R. Jia, Y. Xie, J. Huang, and T. Nakajima, 2015: Modeling study on the transport of summer dust and anthropogenic aerosols over the Tibetan Plateau. Atmospheric Chemistry and Physics, 15(21), 12581–12594, doi:10.5194/acp-15-12581-2015.
**Response: Thanks for your consideration on this points. We have compared the difference among satellite retrieved results, active remote sensing and model simulation in Page 2, Line 5-9. Suggested reference are accepted.**

**Reviewer# 3:**
1.   Page 1, Line 23, "Yunan Province" should be revised as "Yunnan Province".
**Response: Thanks for your carefully check on our manuscript. We have revised the "Yunan Province" into "Yunnan Province" in Page 1, Line 23 and Page 33, Line 8.**

2.   In the title of Figure 2, how did the AODs divide into 50 bins, please specify it for future readers.
**Response: We are sorry for a mistake in the number of bins. In fact, we separate the matchup pairs into 100 bins (not 50 bins) along with AERONET AOD values to obtain finer result. Then in each bins, we draw the bias boxplot and mean bias. Corresponding description is added in the title of Figure 2.**

3.   In Figure 13, 14, and 16, the sudden change of AODs distribution related to MAIAC at the 30°N can be seen clearly. It doesn't make sense. I guess that there seems to be something wrong in the data processing or whatever, please check the MAIAC dataset and specify it in the context.

**Response: The different aerosol model used in MAIAC algorithm for regions higher than 30°N and lower than 30°N is the reason for sudden change of AODs distribution. This is a limitation on MAIAC algorithm and firstly reported in Lyapustin et al., 2018. In our evaluation results, we also find this problem. The reason for the sudden change of AODs distribution are shown in Page 29, Line 17~18.**

4.   The reference of Matins et al. (2017), mainly focused on validation of MAIAC product over South America, was cited frequently in this paper. Such as, the selection of spatial and temporal window, the definition of land cover type of AERONET sites, the strategy of comparison of retrieval accuracy and so on. Furthermore, All these seem to imply us that 1) the method of Matins et al. (2017) is superior or the most reasonably, 2) the difference between this paper and Matins et al. (2017) is only at location of the area of interest. So the authors should make a reasonable evaluation to the reference in the context. Furthermore, the reference of Zhang et al. (2019) in Page 2 Line 25 should be explained the relative simplicity of the assessment methods with more words, which will further demonstrate the comprehensive of the methodology used by the authors.

**Response: Thanks for your suggestion! In fact, method used in Matins et al. (2017) is also applied in other validation studies. In the manuscript, we saved the reference of Matins et al. (2017) in the selection of spatial and temporal window and replace the Matins et al. (2017) reference in other validation method (e.g. Page 10, Line 3-4; Page 12, Line 8; Page 19, Line 4). In comparison with the work of Matins et al. (2017), we also evaluate more aspects than theirs work, e.g. view geometry dependency analysis, spatiotemporal retrieval accuracy analysis and spatiotemporal completeness analysis, etc.**

**            The validation approach used in Zhang et al. (2019) are added Page 2, Line 28-30.**

5.   Page 10, Line 5, the different definition of EE envelope can be found in different reference, how did you consider it and please specify it in the context.

**Response: Commonly used EE envelopes may be $\pm$(0.05+0.15×AOD) and $\pm$(0.05+0.2×AOD). We select more strict EE envelopes $\pm$(0.05+0.15×AOD) in our studies. The EE envelope used in our study can be found in Page 10, Line 4.**